# TGF-β1 secreted by Tregs in lymph nodes promotes breast cancer malignancy via up-regulation of IL-17RB

Shih-Chia Huang[1], Pei-Chi Wei[1], Wendy W Hwang-Verslues[1], Wen-Hung Kuo[2], Yung-Ming Jeng[3], Chun-Mei Hu[1], Jin-Yuh Shew[1], Chiun-Sheng Huang[2], King-Jen Chang[2], Eva Y-HP Lee[1,4] & Wen-Hwa Lee[1,5,*] [iD]

## Abstract

Lymph node (LN) metastasis is commonly associated with systemic distant organ metastasis in human breast cancer and is an important prognostic predictor for survival of breast cancer patients. However, whether tumor-draining LNs (TDLNs) play a significant role in modulating the malignancy of cancer cells for distant metastasis remains controversial. Using a syngeneic mouse mammary tumor model, we found that breast tumor cells derived from TDLN have higher malignancy and removal of TDLNs significantly reduced distant metastasis. Up-regulation of oncogenic Il-17rb in cancer cells derived from TDLNs contributes to their malignancy. TGF-β1 secreted from regulatory T cells (Tregs) in the TDLNs mediated the up-regulation of Il-17rb through downstream Smad2/3/4 signaling. These phenotypes can be abolished by TGF-β1 neutralization or depletion of Tregs. Consistently, clinical data showed that the up-regulation of IL-17RB in cancer cells from LN metastases correlated with the increased prevalence of Tregs as well as the aggressive growth of tumors in mouse xenograft assay. Together, these results indicate that Tregs in TDLNs play an important role in modulating the malignancy of breast cancer cells for distant metastasis. Blocking IL-17RB expression could therefore be a potential approach to curb the process.

**Keywords** breast cancer; IL-17RB; regulatory T cell; TGF-β1; tumor-draining lymph node

**Subject Categories** Cancer; Immunology

## Introduction

Cancer metastasis is a complex process involving tumor microenvironment and systemic changes (Joyce & Pollard, 2009). It has been reported that both blood circulation and lymphatic system can mediate systemic metastasis in human carcinoma (Nathanson *et al*, 2015). In breast cancer, mounting evidence indicates that the initial sites of metastasis are the regional lymph nodes (LNs), while spread of cancer cells goes through lymphatic system (Leong, 2004; Viehl *et al*, 2011; Podgrabinska & Skobe, 2014). Clinical studies have demonstrated a highly significant association between LN metastasis and distant metastasis in breast cancer patients (Abner *et al*, 1998; Rouzier *et al*, 2002; Nathanson *et al*, 2009). The increase in lymphangiogenesis, up-regulation of chemokines and cytokines, and remodel of high endothelial venules are observed in the TDLNs which facilitates cancer cell entry into lymphatic system (Pereira *et al*, 2015). While LNs are initial metastatic sites of breast cancer, the biological influence on cancer cells in the LN microenvironment remains elusive.

It is noted that the LNs could induce anti-tumor immune response to retard tumor spread (Kim *et al*, 2006). Tumor-specific T cells are activated by antigen-presenting dendritic cells (DCs) in the LNs which are able to restrict metastatic outgrowth of disseminated cancer cells (Chamoto *et al*, 2006; Eyles *et al*, 2010). However, in breast cancer patients, altered compositions of immune cells and decreased anti-tumor immune response in TDLNs have been reported (Kohrt *et al*, 2005; Matsuura *et al*, 2006). For example, recruitment of immunosuppressive cells such as regulatory T cells (Tregs), which suppress cytotoxic CD8$^+$ T cells, has been observed in LNs with metastasized cancer cells (Mansfield *et al*, 2009; Nakamura *et al*, 2009; Faghih *et al*, 2014). Moreover, alteration of immune cell profile in the TDLNs has been suggested to be a good predictor of relapse-free and disease-free survival in early stage breast cancer patients (Kohrt *et al*, 2005; Nakamura *et al*, 2009). In

1   Genomics Research Center, Academia Sinica, Taipei, Taiwan
2   Department of Surgery, National Taiwan University Hospital, Taipei, Taiwan
3   Department of Pathology, National Taiwan University Hospital, Taipei, Taiwan
4   Department of Biological Chemistry, University of California, Irvine, CA, USA
5   Institute of New Drug Development, China Medical University, Taichung, Taiwan
   *Corresponding author. Tel: +886 2 27898777; E-mail: whlee@uci.edu

    

addition to the change in immune cell composition, elevated levels of immunosuppressive cytokines such as TGF-$\beta$1, IL-10, and GM-CSF have been observed in the TDLNs of patients with breast cancer and other carcinoma diseases (Dalal *et al*, 1993; Leong *et al*, 2002; Lee *et al*, 2005). These immunosuppressive cytokines may increase Treg differentiation and influence DC maturation in the TDLNs (Munn & Mellor, 2006). Alteration of immune cell composition and enrichment of immunosuppressive cytokines in the TDLNs implicate that a permissive microenvironment is created for cancer cell to survive and expand (Swartz & Lund, 2012).

Evidences from clinical trials suggest that metastasis to LNs reflects tumor aggressiveness but not distant organ metastasis (Gervasoni *et al*, 2007; Leong & Tseng, 2014). Such argument, however, cannot explain the strong association between LN metastasis and distant metastasis in breast cancer patients (Rouzier *et al*, 2002; Ran *et al*, 2010). The experimental mouse model demonstrated that the stimulation of lymphangiogenesis by breast cancer cells secreted growth factors is important for cancer metastasis to distant organ metastasis (Hirakawa *et al*, 2007; Wang *et al*, 2012). Thus, how LNs modulate cancer cells gaining malignancy to metastasize to distant organs remains to be resolved.

In this communication, we found that in the syngeneic mouse mammary tumor models, breast cancer cells derived from the TDLNs exhibited higher malignancy assayed by tumorigenic and metastatic activities. Furthermore, distant metastasis was significantly reduced when the TDLNs were removed at early time point. Elevated prevalence of Tregs in the TDLNs had a prominent effect on promoting cancer malignancy. The enhancement of cancer malignancy was due to the up-regulation of an oncogenic receptor, Il-17rb, by Treg-secreted TGF-$\beta$1. Blocking this TGF-$\beta$1/TGFR paracrine signaling abolished Il-17rb induction as well as inhibited tumorigenic activities of cancer cells. These observations were validated using a cohort of human breast cancer cells derived from either primary tumors or LN metastasis of the same patients, supporting a significant role of TDLNs in modulating cancer cells gained malignancy.

# Results

### Breast cancer cells derived from the TDLNs acquire aggressive phenotypes

The mouse mammary carcinoma 4T1 cells, derived from a spontaneous mammary tumor in BALB/c mice, have high inguinal LN metastatic ability when injected into the inguinal mammary fat pad of the syngeneic mice (Aslakson & Miller, 1992). We employed this syngeneic mouse breast tumor model, which has intact lymphoid organs and immune systems, to examine how TDLNs play for systemic metastasis. 4T1 cells transduced with green fluorescent protein (GFP) and luciferase by lentivirus were used for following allograft assays. A time-course allograft experiment indicated that GFP$^+$ 4T1 cells invaded the inguinal LNs within the first week after injection (Fig 1A). Interestingly, the numbers of invaded GFP$^+$ 4T1 cells slightly reduced in the second week, but increased in the third and fourth week at inguinal LNs (Fig 1B). Compared to LN metastasis, distant organ metastasis was occurred in 4T1-injected mice at the 5 week after injection (Appendix Fig S1). Since 4T1 cells are

resistant to 6-thioguanine (6-TG) (Heppner *et al*, 2000), which allows us to isolate viable cancer cells in either the inguinal LN (4T1$_{LN}$) or the primary tumor (4T1$_{PT}$) from other cell types. We collected 4T1$_{LN}$ at different time points after fat pad injection, and compared their tumorigenic activities with 4T1$_{PT}$. Using soft-agar colony formation assays, we found no significant difference in the colony-forming ability between 4T1$_{LN}$ and 4T1$_{PT}$ collected within the first week after injection. However, a significant enhancement of colony formation in the second and third week of 4T1$_{LN}$ was observed (Fig 1C). Consistent with these results, the size of tumors derived from 4T1$_{LN}$ collected in the first 2 weeks showed no difference compared to the 4T1$_{PT}$-derived tumors (Fig 1D and E). However, tumors derived from 4T1$_{LN}$ collected in the third week were fourfold larger than the tumors derived from 4T1$_{PT}$ in a mouse allograft assay (Fig 1D and E). Similar results were also observed in another syngeneic mammary tumor cell line, EMT6 (Tadmor *et al*, 2011; Fig EV1A–C). To examine whether 4T1$_{LN}$ has greater malignancy potential, we first performed tail-vein injection experiments using BALB/c mice to examine the lung colonization ability. The number of pulmonary tumor nodules derived from 4T1$_{LN}$ was 10-fold greater than that in mice injected with 4T1$_{PT}$ (Fig 1F and G). These results were also observed in another independent tail-vein injection experiment when mice were injected with EMT6$_{LN}$ cells (Fig EV1D and E). Next, we injected 4T1$_{LN}$, 4T1$_{PT}$, EMT6$_{LN}$ and EMT6$_{PT}$ cells into both side of the fourth mammary fat pads of immunodeficient NSG mice. We monitored the spontaneous distant organ metastasis by IVIS imaging. As shown in Fig 1H and I, a higher percentage of distant organ metastasis in 4T1$_{LN}$- or EMT6$_{LN}$-injected NSG mice compared to 4T1$_{PT}$- or EMT6$_{PT}$-injected NSG mice was observed. These data indicated that breast cancer cells isolated from TDLNs have aggressive phenotypes, suggesting that TDLNs play a promoting role to enhance breast cancer malignancy.

### Removal of tumor-draining LN reduces metastasis to distant organs

To directly test whether TDLNs play a role for breast cancer cells spread, we performed allograft assay and removed the tumor in the fourth mammary fat pad on the right side of the mouse with or without inguinal LN on day 8 after injection (Fig 2A). The distant organ metastasis among these mice was monitored by IVIS image analysis (Fig 2B). Twenty control mice with surgical removal of a primary tumor and 18 experimental mice with surgical removal of both primary tumor and inguinal LN were used in this experiment. One week after tumor resection, ten control mice and five experimental mice had tumor growth at the primary sites (Fig EV2) and were excluded from the experiment because a few residual 4T1 tumor cells may not be completely removed after surgery of the primary tumors. Four weeks post-tumor resection, we took IVIS images and observed a lower percentage (three out of 13 mice) of distant organ metastasis in experimental mice comparing with the control mice (five out of 10 mice) (Fig 2C–E). To further demonstrate the role of TDLNs in distant organ metastasis of breast cancer cells, we performed another experiment with a different lymph node surgical removal schedule. For each experimental mouse, the inguinal lymph node in the fourth mammary fat pad on the right side of the mouse was removed 2 weeks before injecting the 4T1 cells in the same fat pad. A sham surgery was performed on the control mice at the same

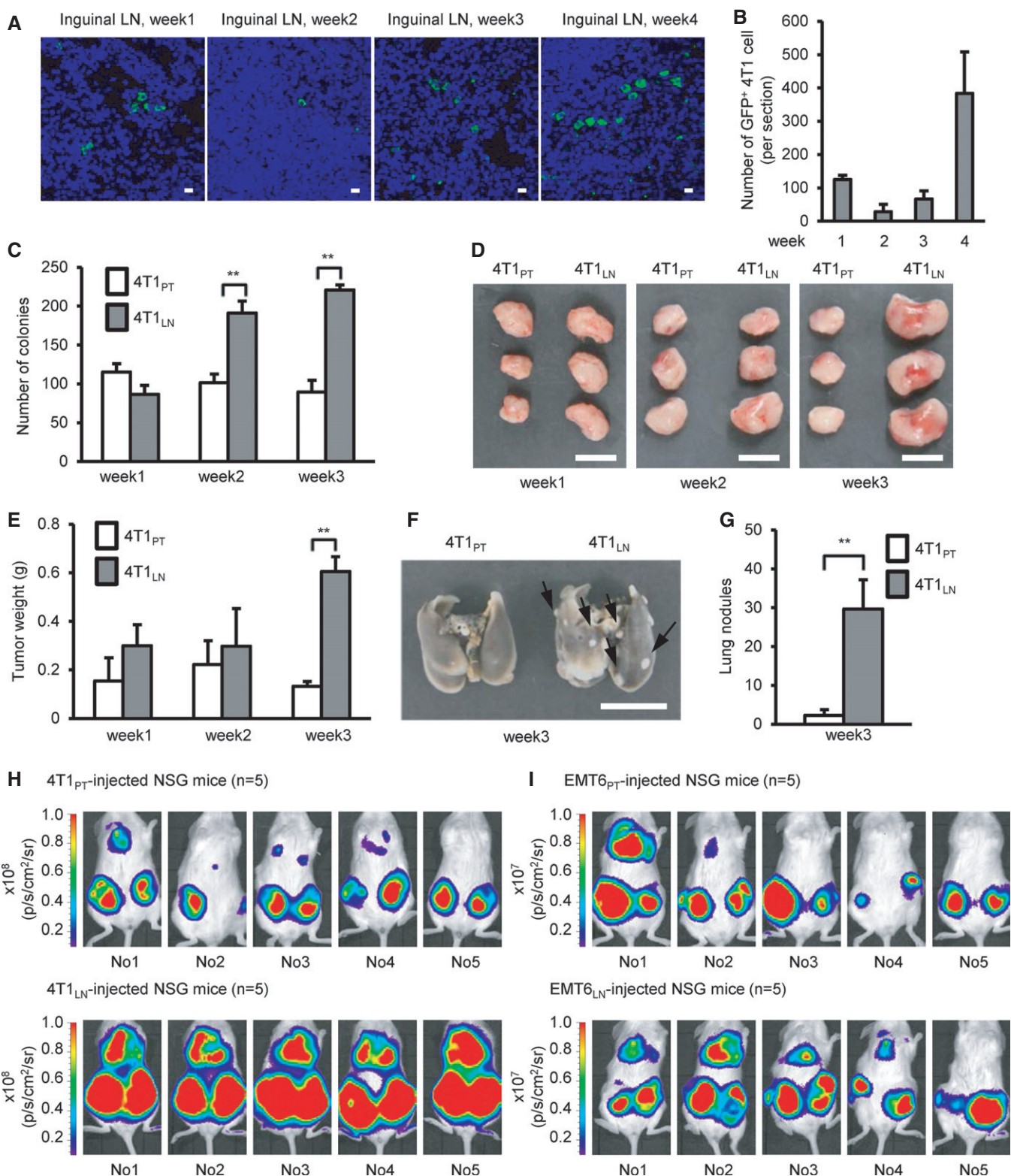

Figure 1.

schedule. Eight days after 4T1 injection, we removed the primary tumors from these mice and then monitored distant organ metastasis by IVIS imaging (Fig 2F). The mice with tumor recurrence at the primary sites after tumor removal were excluded (Fig EV3). Consistently, a relatively lower percentage (one out of 9 mice) of distant organ metastasis was observed in the experimental mice comparing

**Figure 1.  Breast cancer cells derived from tumor-draining lymph nodes gain aggressive malignancy.**

A, B    Inguinal LNs were harvested from 4T1-injected mice at the indicated week after initial injection. LN tissues were fixed and paraffin-embedded. Immunofluorescence was performed with anti-GFP antibody to visualize GFP$^+$ 4T1 cells. (A) Confocal micrographs of the representative field of the LN section. Scale bar: 10 μm. (B) Histograms illustrated that the dynamic changes of GFP$^+$ 4T1 cells metastasized to the inguinal LN. Cells were counted in whole sections (two non-consecutive sections per LN per mouse). Each group was composed of at least two mice.

C–I    4T1-injected or EMT6-injected mice were sacrificed at the indicated week after initial injection. Collagenase-digested specimens from the primary tumors or inguinal LN tissues were cultured in culture dish for cancer cell enrichment. The detailed process is described in the methods section. After several days selection/culture, viable 4T1$_{PT}$, 4T1$_{LN}$, EMT6$_{PT}$, and EMT6$_{LN}$ cells were used for the following assays. (C) Soft-agar colony-forming activity was examined in 4T1$_{PT}$ and 4T1$_{LN}$ cells (5 × 10$^2$ cells/well, $n$ = 4 wells per group). (D, E) Tumorigenesis assays were determined in BALB/c mice ($n$ = 3 mice per group) orthotopically injected with 5 × 10$^2$ 4T1$_{PT}$ or 4T1$_{LN}$ cells. Tumor mass (D) and tumor weight (E) were measured on day 28. Scale bar: 1 cm. (F, G) BALB/c mice ($n$ = 3 mice per group) were injected with 2 × 10$^5$ 4T1$_{PT}$ or 4T1$_{LN}$ cells via tail vein. Lung colonization was examined by lung morphology (F) and the numbers of tumor nodule (G) on day 21. Scale bar: 1 cm. Black arrow: Lung nodule. (H, I) NSG mice ($n$ = 5 mice per group) were orthotopically injected with 4T1$_{PT}$ (5 × 10$^2$), 4T1$_{LN}$ (5 × 10$^2$), EMT6$_{PT}$ (1 × 10$^2$), or EMT6$_{LN}$ (1 × 10$^2$) cells. Distant organ metastasis was examined by bioluminescent images on day 28. The bioluminescent signal (pseudocolor) was recorded as photons per second per centimeter squared per steradian (p/s/cm$^2$/sr), and the luminescent image was overlaid on the photographic image.

Data information: In (A–G), all experimental data were verified in at least two independent experiments. All values are presented as mean or mean ± SD. In (C), **$P$ = 0.000088 and **$P$ = 0.000004 for the 4T1$_{LN}$ cells at week 2 and week 3, respectively. In (E), **$P$ = 0.000006. In (G), **$P$ = 0.000734. Level of significance was determined using two-tailed unpaired $t$-test.

with the control mice (three out of five control mice; Fig 2G–I). Together, these results suggested that the TDLNs play a significant role in modulating breast cancer cells metastasized to distant organs.

**Oncogenic Il-17rb is up-regulated in cancer cells in the TDLNs**

To determine the underlying mechanism by which breast cancer cells acquired aggressive malignant phenotypes in the LNs, we performed cDNA microarray analysis to explore the differential gene expression between 4T1$_{LN}$ and 4T1$_{PT}$. 4T1$_{LN}$ and 4T1$_{PT}$ were collected from BALB/c mice in the third week after injection. Compared to 4T1$_{PT}$, 65 genes were expressed at least log$_2$ 2-fold higher in 4T1$_{LN}$ (Table EV1). Among these genes, we were particularly interested in five genes encoding cell surface proteins as they were most likely to be involved in the interaction between cancer cells and the microenvironment of LNs or primary tumor sites (Fig 3A). We used RT–qPCR to confirm the results of the cDNA microarray and found that Il-17rb, Gpr56, and Scara5 were significantly up-regulated in 4T1$_{LN}$ when compared to 4T1$_{PT}$ (Fig 3B). Consistent with gene expression, protein level of Il-17rb and Gpr56, but not Scara5, was up-regulated in 4T1$_{LN}$ (Fig 3C and D). To further examine whether these three genes were associated with the aggressive malignant phenotypes observed in 4T1$_{LN}$, *Il-17rb*, *Gpr56*, and *Scara5* were depleted in 4T1 cells individually using a lentiviral shRNA system (Fig 3E). These 4T1 cells were then subjected to soft-agar colony-forming assays. The colony-forming ability was significantly suppressed only in *Il-17rb*-knockdown cells, not in *Gpr56*- or

*Scara5*-knockdown cells (Fig 3F). Similarly, Il-17rb was up-regulated in EMT6$_{LN}$ when compared to EMT6$_{PT}$ cells (Appendix Fig S2). Consistent with the previous report (Huang *et al*, 2014), over-expression of Il-17rb in 4T1 cells significantly enhanced the colony-forming ability (Fig 3G and H). To further address whether Il-17rb contributes to tumor growth and metastasis, we used a RNA-guided CRISPR-Cas9 system to delete Il-17rb in 4T1 cells for *in vivo* tumor growth and lung colonization assays (Fig 3I). Both tumor growth and lung nodules were reduced in *Il-17rb*-knockout 4T1 cells (Fig 3J–M). These results suggested that up-regulation of *Il-17rb* contributes to the aggressive malignancy phenotypes of 4T1$_{LN}$ cells.

**Tregs in the TDLNs secrete factors to induce Il-17rb expression**

To investigate how *IL-17rb* expression was induced at the site of TDLN, we established an *in vitro* 5-day transwell co-culture system using 4T1 cells cultured in the bottom well and total cells collected from LNs cultured in the inserts (Fig 4A). The cells from the TDLNs were prepared from tumor-bearing BALB/c mice at different time points post fat pad injection (wk1, wk2, and wk3). Cells isolated from the LNs of un-injected mice were used as a control. In this experiment, the gene and protein expression of *Il-17rb* in 4T1 cells was increased when co-cultured with cells from TDLNs and reached the highest level when co-cultured with TDLN cells isolated in week 3 postinjection (Fig 4B and C). Consistent with the induction of Il-17rb, the colony-forming ability of the co-cultured 4T1 was also increased and reached the highest level after co-cultured with LN cells isolated in week 3 postinjection (Fig 4D). These results

**Figure 2.  Removal of tumor-draining lymph node reduces distant organ metastasis of breast cancer cells.**

A, B    Schematic diagram of tumor resection with or without tumor-draining lymph node removal in 4T1 tumor-bearing mice.

C, D    Distant organ metastasis was examined by bioluminescent images of 4T1 tumor-bearing mice after tumor resection with ($n$ = 13 mice) or without ($n$ = 10 mice) tumor-draining lymph node dissection on week 5. The bioluminescent signal (pseudocolor) was recorded as photons per second per centimeter squared per steradian (p/s/cm$^2$/sr), and the luminescent image was overlaid on the photographic image. For the additional details, see the Materials and Methods section.

E    Summary table of distant organ metastasis derived from each surgery group in (C, D).

F    Schematic diagram of tumor resection in inguinal lymph node pre-removed or sham control 4T1 tumor-bearing mice.

G, H    Distant organ metastasis was examined by bioluminescent images of inguinal lymph node pre-removed tumor-bearing mice ($n$ = 9 mice) or sham surgery tumor-bearing mice ($n$ = 5 mice) after tumor resection on week 5. The bioluminescent signal (pseudocolor) was recorded as photons per second per centimeter squared per steradian (p/s/cm$^2$/sr), and the luminescent image was overlaid on the photographic image. For additional details, see the Materials and Methods section.

I    Summary table of distant organ metastasis derived from each surgery group in (G, H).

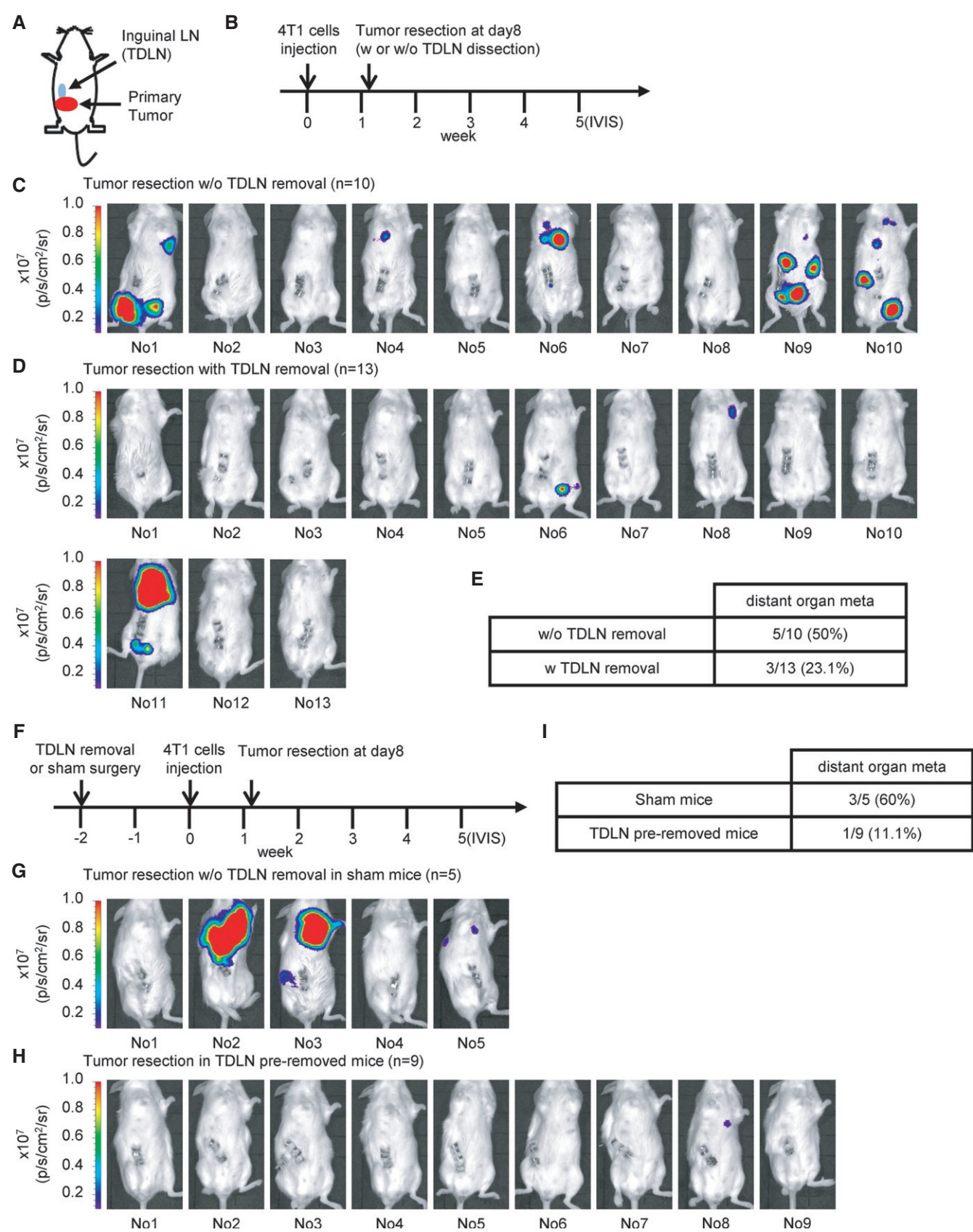

**Figure 2.**

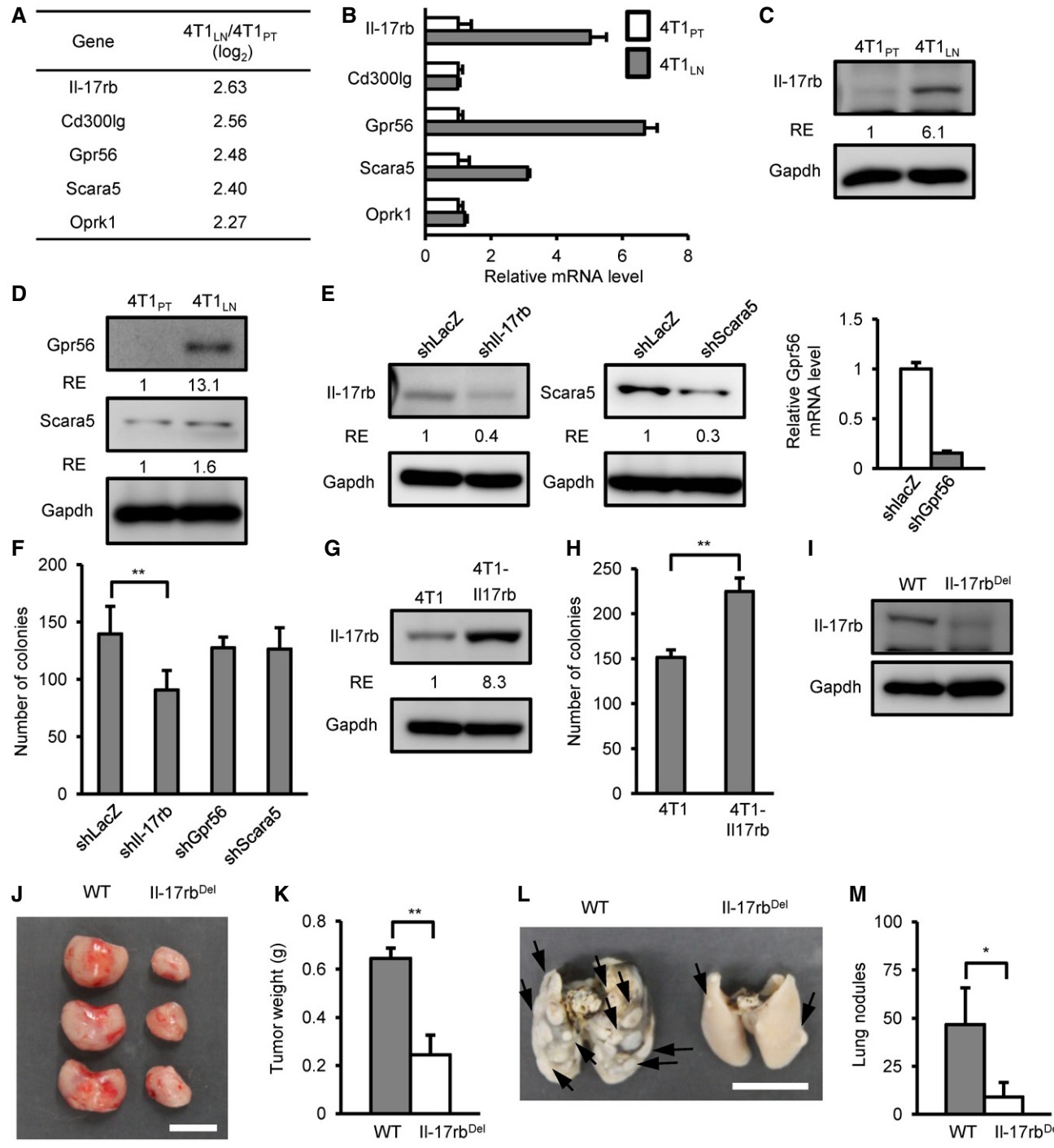

Figure 3.

suggested that factors secreted from cells of the TDLNs are responsible for the induction of Il-17rb expression, which attributes to the enhancement of colony-forming activity in breast cancer cells.

To explore which subset of cells in the TDLNs mediated *Il-17rb* up-regulation in cancer cells, we isolated individual subset of immune cells by FACS sorter for performing the co-culture experiment using 4T1 cells as described above. When 4T1 cells were co-cultured only with CD4[+] T-cell subset, but not with other subsets, Il-17rb expression was significantly induced (Fig 4E and F). Among

CD4[+] T-cell subpopulations, increased prevalence of Tregs has been reported in the TDLNs in breast cancer patients (Mansfield *et al*, 2009; Nakamura *et al*, 2009). Similar observation has also been reported in the TDLNs in a 4T1 syngeneic mouse breast tumor model (Darrasse-Jeze *et al*, 2009). Consistently, a significant increase in Tregs in the TDLNs was observed in the second and third week after 4T1 cell injection (Fig 4G, left panel). Increase of Tregs in the TDLNs may come from spleen and peripheral blood through circulation (Fig 4G, middle and right panels), or from the

Figure 3.  Up-regulation of Il-17rb contributes to the aggressive malignancy phenotypes of breast cancer cell derived from tumor-draining lymph node.

A    Gene expression profiles were shown at $4T1_{LN}$ to $4T1_{PT}$ cells. Five genes encoding cell surface proteins were identified among up-regulated genes.

B    mRNA expression of each candidate gene in $4T1_{PT}$ and $4T1_{LN}$ cells was determined by RT–qPCR. Gapdh was used as an internal control.

C, D  Il-17rb, Gpr56, and Scara5 expression in $4T1_{PT}$ and $4T1_{LN}$ cells were examined by Western blotting analysis.

E    Western blotting and RT–qPCR analysis of Il-17rb, Gpr56, and Scara5 expression in 4T1 cells transduced with Il-17rb, Gpr56, Scara5, or control LacZ shRNA lentivirus, respectively.

F    Soft-agar colony-forming activity was examined in lentivirus-transduced shIl-17rb, shGpr56, shScara5, or shLacZ 4T1 cells ($5 \times 10^2$ cells/well, $n = 6$ wells per group).

G    Western blotting analysis was used to detect ectopic expression of Il-17rb in 4T1 cells infected with either lentiviruses carrying Il-17rb cDNA or empty vector.

H    Soft-agar colony-forming activity was examined using control or Il-17rb over-expressing 4T1 cells ($5 \times 10^2$ cells/well, $n = 6$ wells per group).

I    Western blotting analysis was used to examine Il-17rb in a representative Il-17rb$^{Del}$ 4T1 clone derived from Il-17rb-knockout 4T1 cells using a CRISPR/Cas9 system.

J, K  Tumorigenesis assays were determined in BALB/c mice ($n = 3$ mice per group) orthotopically injected with $5 \times 10^2$ 4T1 cells from WT or Il-17rb$^{Del}$ clone. Tumor mass (J) and weight (K) were measured on day 28. Scale bar: 1 cm.

L, M  BALB/c mice ($n = 3$ mice per group) were injected with $5 \times 10^5$ 4T1 cells from WT or Il-17rb$^{Del}$ clone via tail vein. Lung colonization was examined by lung morphology (L) and the numbers of tumor nodule (M) on day 21. Scale bar: 1 cm. Black arrow: Lung nodule.

Data information: All experimental data were verified in at least two independent experiments. All values are presented as mean ± SD. In (B), data were presented as means ± SD (triplicate measurement). In (C, D, E, G, and I), the intensity of each band was quantified using the ImageJ software, and Gapdh was used as a loading control. Relative expression (RE) of protein levels in each figure is indicated. In (F), **$P = 0.00216$ for the shIl-17rb 4T1 cells. In (H), **$P = 0.0000009$. In (K), **$P = 0.00012$. In (M), *$P = 0.032862$. Level of significance was determined using two-tailed unpaired $t$-test.

Source data are available online for this figure.

conversion of Tregs from resting CD4$^+$ T cells (Olkhanud et al, 2011). Tregs isolated from TDLNs of 4T1-injected mice remain exhibit similar T-cell suppressive activity to Tregs isolated from un-injected mice (Appendix Fig S3). Since Th17 cells were increased in the blood, spleen, and primary tumor in 4T1-injected mice (Qian et al, 2013), the population of RORγt$^+$ Th17 cells or IL-17A$^+$ CD4$^+$ T cells in TDLNs was also examined by FACS analysis. However, in comparison with CD4$^+$Foxp3$^+$ Tregs, a relatively lower percentage of RORγt$^+$ Th17 cells or IL-17A$^+$ CD4$^+$ T cells were observed in the TDLNs in the third week after 4T1 cell injection (Appendix Fig S4). Thus, it is likely that Tregs in the TDLNs may be responsible for the induction of Il-17rb in cancer cells. To test this, CD4$^+$CD25$^+$ Tregs were sorted from the TDLNs 3 weeks postinjection and then co-cultured with the 4T1 cells. In this co-culture setting, the expression of *Il-17rb* in 4T1 cells was significantly induced (Fig 4H). Further analysis of CD4$^+$ T-cell subpopulations revealed that CD4$^+$CD25$^-$ effector T cells were not able to induce *Il-17rb* expression of 4T1 cells (Fig 4H). Interestingly, the total population of CD4$^+$ T cells had the highest induction activity (Fig 4H), suggesting that other

CD4$^+$ non-Treg cells in the TDLNs may indirectly participate the induction of Il-17rb in cancer cells.

## Depletion of Tregs abolishes Il-17rb induction and enhanced malignancy in breast cancer cells

To further affirm that Tregs in the TDLNs are responsible for promoting cancer cell malignancy, we depleted Tregs in the 4T1-injected mice using anti-CD25 antibody clone PC61, which has been widely used to deplete Tregs for characterizing Treg function *in vivo* (Setiady et al, 2010). However, it is reported that tumor-evoked regulatory B cells (Bregs; Olkhanud et al, 2011) also express CD25$^+$ and enhance breast cancer lung metastasis in 4T1 mouse model (Olkhanud et al, 2009). To precisely pinpoint which cell type is responsible for this phenotype, we depleted specific cell type by injecting anti-CD4 Ab to deplete all CD4$^+$ T cells including Foxp3$^+$ Tregs, anti-CD25 Ab to deplete Foxp3$^+$ Tregs including CD25$^+$ Bregs, and control IgG, into 4T1-injected mice. We then first examined the composition of immune cells after these antibody

Figure 4.  Tregs in the tumor-draining lymph node microenvironment mainly contribute to the up-regulation of Il-17rb in breast cancer cells.

A    Schematic diagram of the *in vitro* co-culture system using 4T1 cells and total cells isolated from tumor-draining lymph nodes.

B, C  4T1-injected BALB/c mice were sacrificed at the indicated week after initial injection. Total cells isolated from inguinal lymph node tissues were transwell co-cultured with 4T1 cells. Inguinal lymph node tissues came from un-injection BALB/c mice as control. After 5-day co-culture, 4T1 cells at lower well were examined in the RT–qPCR (B) or Western blotting (C) analyses of Il-17rb expression. Gapdh was used as an internal control or as a loading control.

D    Soft-agar colony-forming activity was examined using co-cultured 4T1 cells at lower well ($5 \times 10^2$ cells/well, $n = 4$ wells per group).

E, F  4T1-injected BALB/c mice were sacrificed at 3 weeks postinitial injection. Each lymphocyte subsets isolated from inguinal lymph node tissues by FACS sorting were transwell co-cultured with 4T1 cells. After 5-day co-culture, 4T1 cells at lower well were used for the RT–qPCR (E) or Western blotting (F) analyses of Il-17rb expression. Gapdh was used as internal control or as a loading control.

G    Percentage of CD4$^+$Foxp3$^+$ Tregs of total CD4$^+$ cells in inguinal LN, spleen, and peripheral blood of BALB/c mice ($n = 3$ mice per group) injected with 4T1 (Blue) or γ-irradiated 4T1 (Red) cancer cells was analyzed by FACS.

H    4T1 cells co-cultured with the indicated lymphocyte populations isolated from inguinal lymph node tissues of 4T1 tumor-bearing mice by FACS sorting. After 5-day co-culture, 4T1 cells at lower well were analyzed for Il-17rb expression by RT–qPCR analysis. Gapdh was used as internal control.

Data information: All experimental data were verified in at least two independent experiments. All values are presented as mean ± SD. In (B, E, H), values were presented as mean ± SD of triplicate measurement. In (C, F), the intensity of each band was quantified using the ImageJ software. Relative expression (RE) of Il-17rb levels in each figure is indicated. In (D), **$P = 0.0069$ and **$P = 0.000015$ for the week 2 LNs co-cultured 4T1 cells and week 3 LNs co-cultured 4T1 cells, respectively. In (G), **$P = 0.000233$ and **$P = 0.001717$ for the percentage of CD4$^+$Foxp3$^+$ Tregs at week 2 and week 3, respectively. Level of significance was determined using two-tailed unpaired $t$-test.

Source data are available online for this figure.

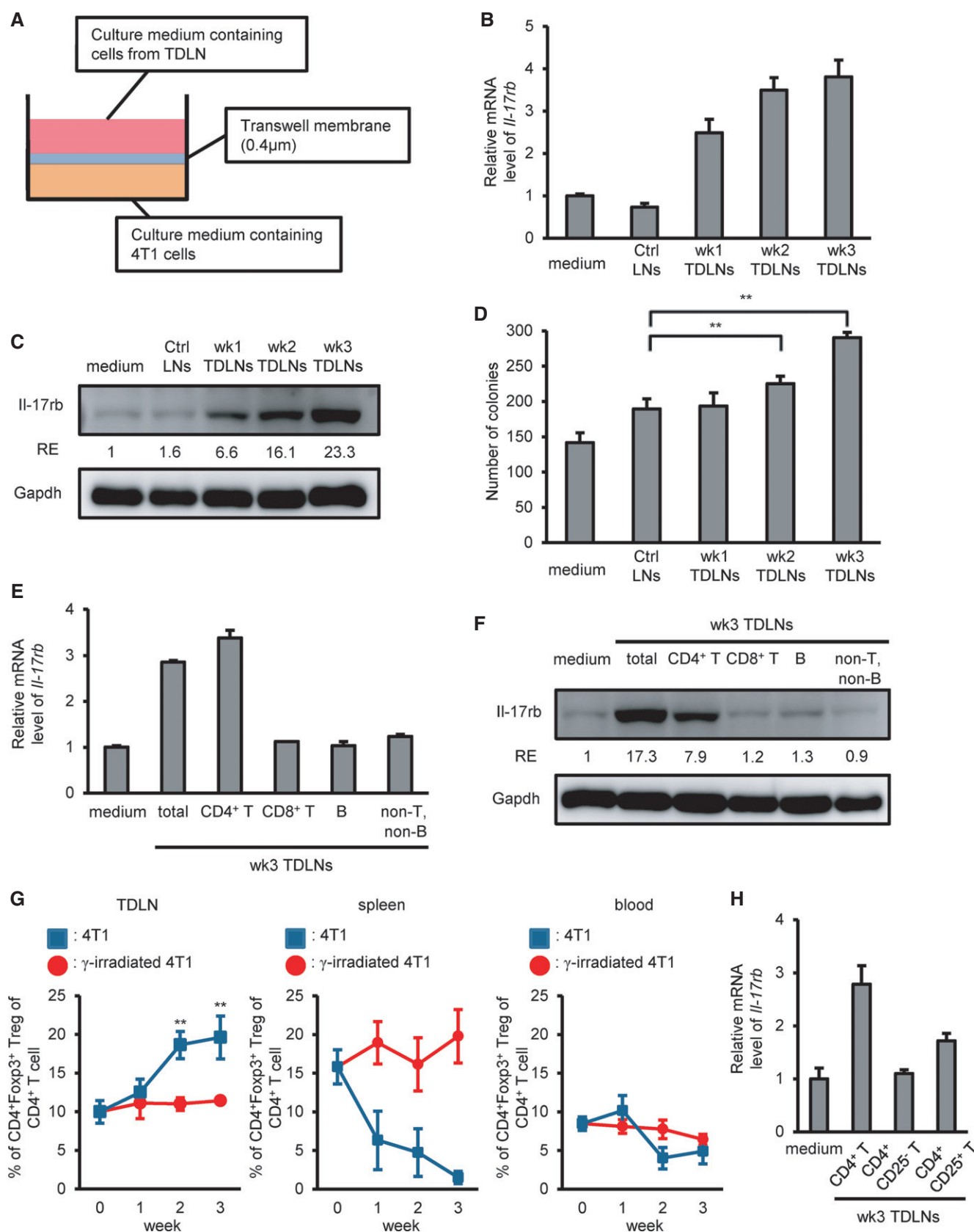

**Figure 4.**

treatments by FACS analysis. We found a significant reduction in the Treg population of the TDLNs derived from CD4$^+$ T cell-depleted or CD25$^+$ cell-depleted mice (Fig 5A). Compared to the significant reduction in Treg population, a great proportion of Bregs was still observed in the TDLNs derived from either CD4$^+$ T cell-depleted or CD25$^+$ cell-depleted mice (Fig EV4). It was implicated that Breg population may be functional in the TDLNs in both groups of mice. The 4T1$_{LN}$ were then isolated from the CD4$^+$ T cell-depleted, CD25$^+$ cell-depleted or control IgG-treated mice for further analyses. The expression of Il-17rb in the 4T1$_{LN}$ cells collected from either CD4$^+$ T cell-depleted mice or CD25$^+$ cell-depleted mice was significantly diminished (Fig 5B). Moreover, both tumor growth and spontaneous distant organ metastasis of 4T1$_{LN}$ from the CD4$^+$ T cell-depleted mice or CD25$^+$ cell-depleted mice were significantly reduced (Fig 5C–E). However, this inhibition of spontaneous metastasis ability was not found in 4T1$_{PT}$ isolated from CD4$^+$ T cell-depleted or CD25$^+$ cell-depleted mice when compared with 4T1$_{PT}$ isolated from control IgG-treated mice (Fig 5E). Taken together, these results suggest that Tregs in the TDLNs are the major cell type to induce Il-17rb expression in breast cancer cells to enhance malignancy.

### TGF-β1 secreted from Tregs induces the expression of Il-17rb via Smad2/3/4 signaling

The above co-culture results indicated that soluble factors secreted from Tregs were involved in the up-regulation of Il-17rb in the 4T1 cells (Fig 4). It is noted that two well-characterized cytokines, TGF-β1 and IL-10, have been reported to be secreted by Tregs (Jiang & Chess, 2006) and also contribute to tumor promotion in breast cancer (Hamidullah et al, 2012; Zarzynska, 2014). To test which soluble factors were involved in the up-regulation of Il-17rb, we added neutralizing antibodies against TGF-β1 or IL-10 to the Tregs-4T1 co-culture. Treatment with anti-TGF-β1 antibody, but not anti-IL-10 antibody, significantly diminished the up-regulation of Il-17rb expression (Fig 6A). To further confirm this observation, we treated 4T1 cells with recombinant TGF-β1 or IL-10 proteins and found that TGF-β1, but not IL-10, significantly up-regulated Il-17rb gene and protein expression (Fig 6B and C, Appendix Fig S5A and B). Similar results were also obtained when EMT6 cells were used (Appendix Fig S5C and D).

To determine whether the TGF-β1/TGFBR1 signaling was required for Il-17rb up-regulation, we depleted Tgfbr1 by a lentiviral shRNA system. Upon Tgfbr1 depletion, Il-17rb expression was no longer able to be up-regulated by TGF-β1 recombinant protein in 4T1 cells (Fig 6D). To further demonstrate that TGF-β1 plays a critical role in promoting tumorigenic ability of cancer cells in the TDLNs, we performed co-culture experiments using Tgfbr1-depleted or non-depleted 4T1 cells with cells from the TDLNs. Compared to the non-depleted 4T1 cells, the colony-forming ability of Tgfbr1-depleted 4T1 cells could not be enhanced by the TDLNs (Fig 6E).

To demonstrate that TGF-β1 signaling results in up-regulation of Il17rb, we first analyzed the sub-cellular localization of SMADs in 4T1 cells upon TGF-β1 treatment using a commercial sub-cellular protein fractionation kit for cultured cells (Thermo Scientific #78840) to isolated cytoplasm-, nucleoplasm-, and chromatin-binding fractions of TGF-β1-treated 4T1 cells. We found that TGF-β1 treatment induced the chromatin binding of Smad2/3 and Smad4, but

not the inhibitory Smad6 (Fig EV5). We then tested which Smad molecule was necessary for Il-17rb expression by using lentiviruses to deliver Smad2, Smad3, or Smad4 shRNA into 4T1 cell. As shown in Fig 6F–H, TGF-β1-induced Il-17rb expression was abolished in either one of Smad2-, Smad3-, or Smad4-depleted 4T1 cells (Fig 6F–H), consistent with the notion that TGF-β1 signaling resulted in Smad2–Smad4-cofactor or Smad3–Smad4-cofactor complex formation in the nucleus to regulate target gene expression (Massague, 2008).

To demonstrate that the TGF-β1-induced Il-17rb up-regulation contributed to the increase of colony-forming ability, we used Il-17rb-knockdown 4T1 cells for the co-culture assay with cells from the TDLNs. Depletion of Il-17rb abolished the colony-forming ability induced by the TDLNs (Appendix Fig S6). Furthermore, since IL-17B/IL-17RB signaling activates NF-κB in human breast cancer (Huang et al, 2014), NF-κB nuclear translocation can serve as the measurement for the consequence of Il-17rb induction. As shown in Fig 6I and J, upon recombinant mouse IL-17B treatment, a clear nuclear translocation of NF-κB was observed in 4T1 cells pre-treated with TGF-β1 for 5 days (Fig 6I), while this was not the case in both Il-17rb$^{Del}$ 4T1 cells nor in TGF-β1-pre-treated Il-17rb$^{Del}$ 4T1 cells (Fig 6J). Taken together, these results suggested that TGF-β1 secreted from Tregs induces the expression of Il-17rb via Smad2/3/4 signaling and Il-17rb/Il-17b signaling activates NF-κB pathway to contribute to the malignancy of 4T1$_{LN}$ cells.

### High IL-17RB expression in cancer cells at the TDLNs is detected in human clinical specimens

To further investigate whether the Tregs mediated Il-17rb up-regulation in the 4T1 mouse model has clinical significance, we examined the expression of IL-17RB in human breast cancer specimens, including primary tumors and LN metastasis from the same patients by immunohistochemistry (IHC). Among the 76 cases analyzed, 60 cases showed expression of IL-17RB in either primary tumors or LNs. The remaining 16 cases were excluded due to the absence of IL-17RB expression in both primary tumors and LNs. In these 60 cases, the level of IL-17RB was significantly higher in cancer cells in the LNs than those in the primary sites (Fig 7A and B). Consistently, higher percentage of Foxp3$^+$ Tregs was detected in LNs than in primary sites using IHC (Fig 7C). Interestingly, the relationship between IL-17RB expression and the percentage of Tregs was positively correlated (Fig 7D). Therefore, these results further affirmed the observation that IL-17RB expression was up-regulated in cancer cells in the TDLNs.

### Cancer cells isolated from LN metastasis of breast cancer patients exhibit enhanced tumorigenic activity than cells from the primary tumor

Next, we compared the tumorigenic activity of breast cancer cells derived either from patients' LNs or primary sites using a xenograft mouse model. Breast cancer cells freshly isolated from either the primary site (hBC$_{PT}$) or the paired LN metastasis (hBC$_{LN}$) were mixed with Matrigel and orthotopically injected into immunodeficient NOD/SCID/IL2Rγ$^{null}$ mice to examine the tumor outgrowth. These cancer cells were already interacted with their tumor microenvironment either at the primary site or TDLN in the patients for a long time. To avoid graft-versus-host response, CD45$^+$

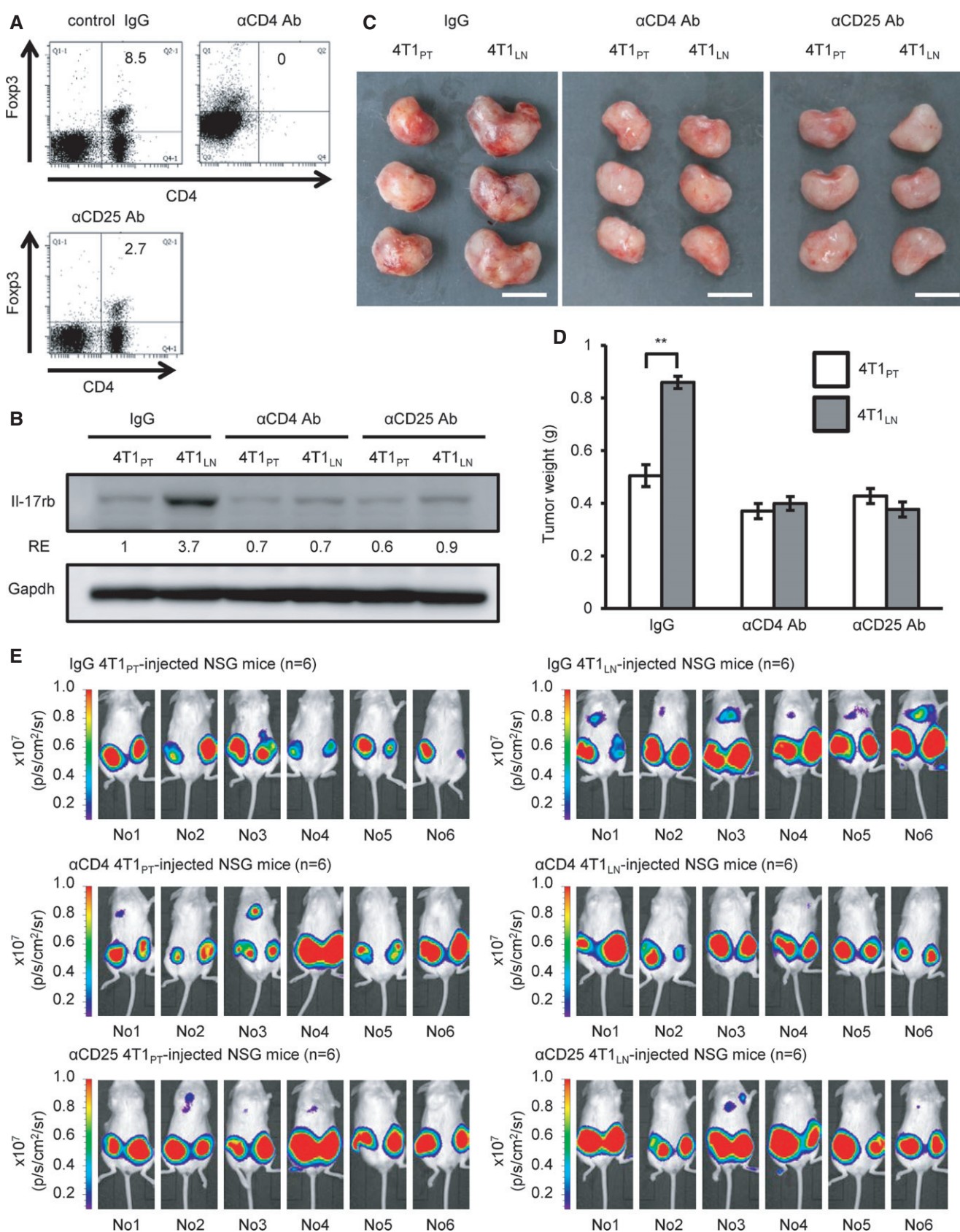

**Figure 5.**

Figure 5.  Depletion of Tregs in the tumor-draining lymph node microenvironment abolishes Il-17rb induction and enhanced malignancy in breast cancer cells.

A    4T1-injected BALB/c mice were treated with control IgG, anti-CD4, or anti-CD25 neutralizing antibody at the day 15 and day 18 after initial injection. Mice were sacrificed at 3 weeks after initial injection. Depletion of $CD4^+Foxp3^+$ Tregs in inguinal LNs was confirmed by FACS analysis.
B    Western blotting analysis of Il-17rb expression was performed in $4T1_{LN}$ or $4T1_{PT}$ cells collected from control IgG-, anti-CD4, or anti-CD25 antibody-treated 4T1 tumor-bearing mice.
C, D    BALB/c mice were orthotopically injected with $5 \times 10^2$ $4T1_{LN}$ or $4T1_{PT}$ cells collected from control IgG-, anti-CD4, or anti-CD25 antibody-treated 4T1 tumor-bearing mice ($n = 3$ mice per group). Tumor mass (D) and tumor weight (E) were measured on day 28. Scale bar: 1 cm.
E    NSG mice ($n = 5$ mice per group) were injected with $5 \times 10^2$ $4T1_{LN}$ or $4T1_{PT}$ cells collected from control IgG-, anti-CD4, or anti-CD25 antibody-treated 4T1 tumor-bearing mice. Distant organ metastasis was examined by bioluminescent images on day 21. The bioluminescent signal (pseudocolor) was recorded as photons per second per centimeter squared per steradian (p/s/cm²/sr), and the luminescent image was overlaid on the photographic image.

Data information: In (A–D), all experimental data were verified in at least two independent experiments. All values are presented as mean ± SD. In (B), the intensity of each band was quantified using the ImageJ software, and Gapdh was used as a loading control. Relative expression (RE) of Il-17rb levels in each sample to $4T1_{PT}$ cells derived from control IgG-treated 4T1 tumor-bearing mice is indicated. In (D), **$P = 0.000209$. Level of significance was determined using two-tailed unpaired $t$-test. Source data are available online for this figure.

hematopoietic cells were depleted before injecting equal numbers of $EpCAM^+$ (epithelial cell surface marker) breast cancer cells into the fourth mammary fat pads (Fig 8A). Six months after injection, tumor growth was evaluated (Fig 8B). Consistent with the findings in the 4T1 mouse breast tumor model, tumors derived from the $hBC_{LN}$ cells were larger than those derived from the $hBC_{PT}$ cells (Fig 8C). The expression of human pan-cytokeratin confirmed that these tumors were originated from human specimens (Fig 8D and E). Importantly, up-regulation of IL-17RB was also observed in the tumors derived from the $hBC_{LN}$ cells, compared to those derived from $hBC_{PT}$ cells (Fig 8F). A positive correlation between tumor growth and the IL-17RB expression in tumors derived from $hBC_{LN}$ was observed (Fig 8G). Since these immunodeficient NOD/SCID/IL2Rγ$^{null}$ mice lack Tregs, the expression of IL-17RB in the $hBC_{PT}$ and the $hBC_{LN}$ cells in these mice must be through non-Tregs mechanism. It is possible that IL-17RB in the $hBC_{LN}$ was up-regulated constantly in human TDLN perhaps through epigenetic machinery. The similar effect has been shown in our previous findings in the fibroblasts co-cultured with breast cancer cells (Tyan et al, 2011, 2012). Thus, when evaluating the expression of IL-17RB in the $hBC_{PT}$ and the $hBC_{LN}$ cells in mice without Tregs, $hBC_{LN}$ still express the high amount of IL-17RB. These results suggested that human breast cancer derived from LNs acquired high tumorigenic activity and that was correlated with the up-regulated IL-17RB.

# Discussion

There has been a long-standing debate over the role of LNs in promoting cancer malignancy. Some clinical evidence has suggested that metastasis to the LNs in breast cancer patients is strongly associated with distant organ metastasis, poor disease-free survival, and shorter overall survival (Rouzier et al, 2002; Ran et al, 2010). Although breast cancer cell-induced lymphangiogenesis in TDLNs is important to distant organ metastasis in mouse model, there is no direct evidence to demonstrate that breast cancer cells metastasized to LNs was required for further distant organ metastasis (Hirakawa et al, 2007; Shibata et al, 2008). Several clinical trials showed no survival benefits for patients underwent lymphadenectomy (Gervasoni et al, 2007). Thus, whether TDLNs involved in the progression of systemic metastasis remain controversial (Ran et al, 2010; Pereira et al, 2015). Using a syngeneic mouse model, we observed that breast tumor cells derived from TDLN gained higher malignancy compared with that from the primary site and removal of TDLNs significantly reduced distant metastasis (Figs 1C–I and 2). Further investigation showed that cancer cells in the TDLNs enhanced their malignancy by up-regulating oncogenic receptor, IL-17rb, in both syngeneic mouse breast tumor model and clinical patients (Figs 3A–C and 7A and B). These results provided an unambiguous evidence to support the importance of TDLNs in promoting distant organ metastasis.

Figure 6.  TGF-β1 secreted from Tregs induces Il-17rb expression of breast cancer cells via Smad2/3/4 signaling.

A    $CD4^+CD25^+$ Tregs were isolated from inguinal LN of BALB/c mice injected with 4T1 cells for 3 weeks and were transwell co-cultured with 4T1 cells in the presence of control IgG or neutralizing antibodies against IL-10 or TGF-β1. After 5-day co-culture, 4T1 cells at lower well were analyzed for Il-17rb expression by RT–qPCR analysis. Gapdh was used as internal control.
B, C    Western blotting analyzed Il-17rb in 4T1 cells treated with recombinant TGF-β1 (B) or IL-10 (C) proteins for 5 days.
D    Western blotting analyzed Il-17rb in shLacZ or shTgfbr1 4T1 cells treated with recombinant TGF-β1 for 5 days.
E    4T1-injected BALB/c mice were sacrificed at 3 weeks postinitial injection. Total cells isolated from inguinal lymph node tissues were transwell co-cultured with 4T1 cells. After 5-day co-culture, soft-agar colony-forming activity was determined in co-cultured shLacZ or shTgfbr1 4T1 cells at lower well. ($5 \times 10^2$ cells/well, $n = 6$ wells per group).
F–H    Western blotting analyzed Il-17rb in shLacZ, shSmad2 (F), shSmad3 (G), or shSmad4 (H) 4T1 cells treated with recombinant TGF-β1 for 5 days.
I, J    WT 4T1 (I) or Il-17rb$^{Del}$ 4T1 (J) cells were treated with or without recombinant TGF-β1 for 5 days. After treatment, nuclear translocation of NF-κB p65 was assayed by Western blotting in cells treated with recombinant mouse IL-17B for 2 h. Nuclear Matrix Protein p84 was used as a loading control of nuclear extracts.

Data information: All experimental data were verified in at least two independent experiments. In (A), values were presented as means ± SD of triplicate measurement. In (B, C, D, F, G, H, I, J), the intensity of each band was quantified using the ImageJ software. Gapdh, α-tubulin, or p84 was used as a loading control. Relative expression (RE) of protein levels in each sample to control 4T1 cells or Il-17rb$^{Del}$ 4T1 clone is indicated. In (E), data were presented as means ± SD. **$P = 0.000615$. Level of significance was determined using two-tailed unpaired $t$-test. Source data are available online for this figure.

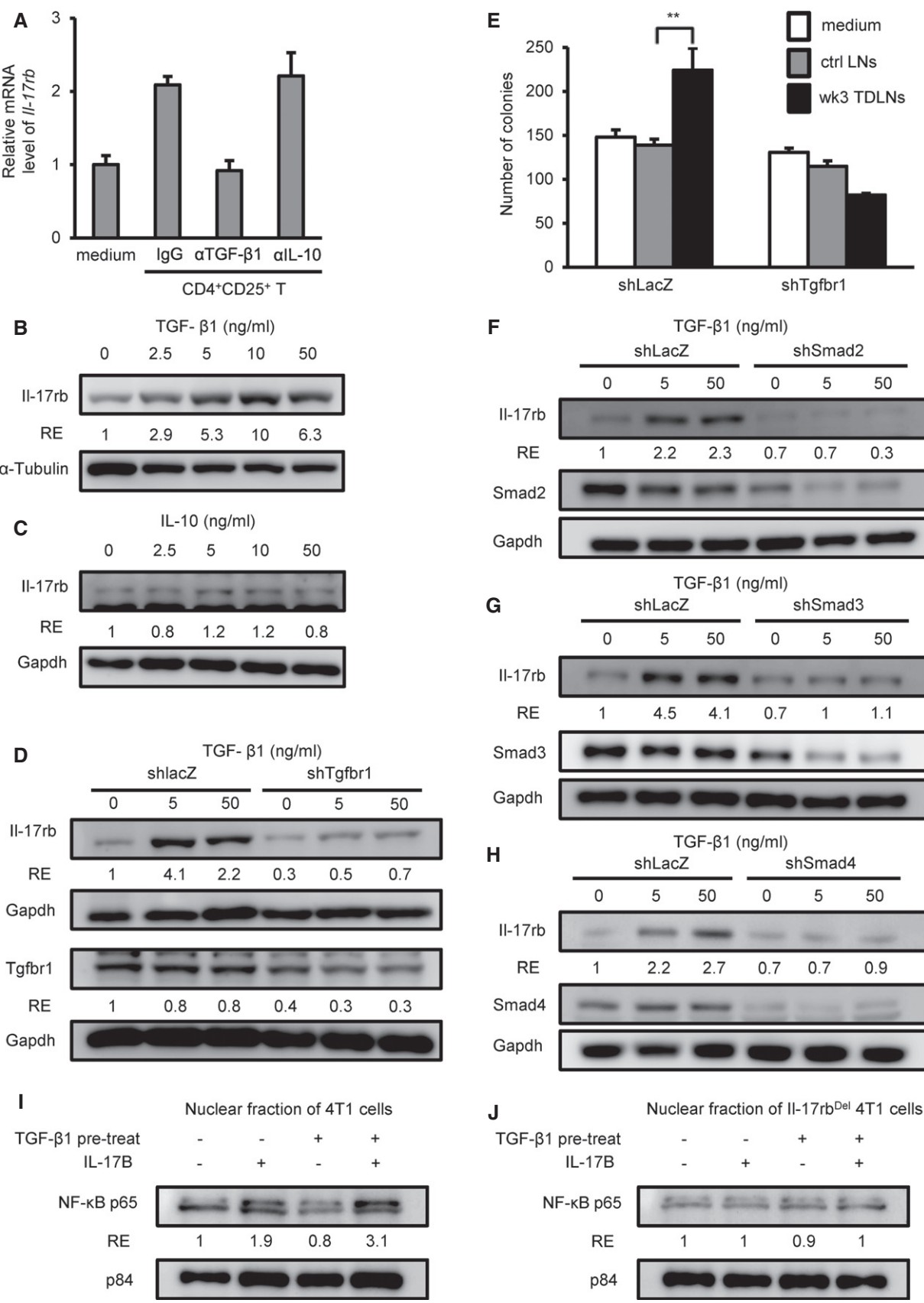

**Figure 6.**

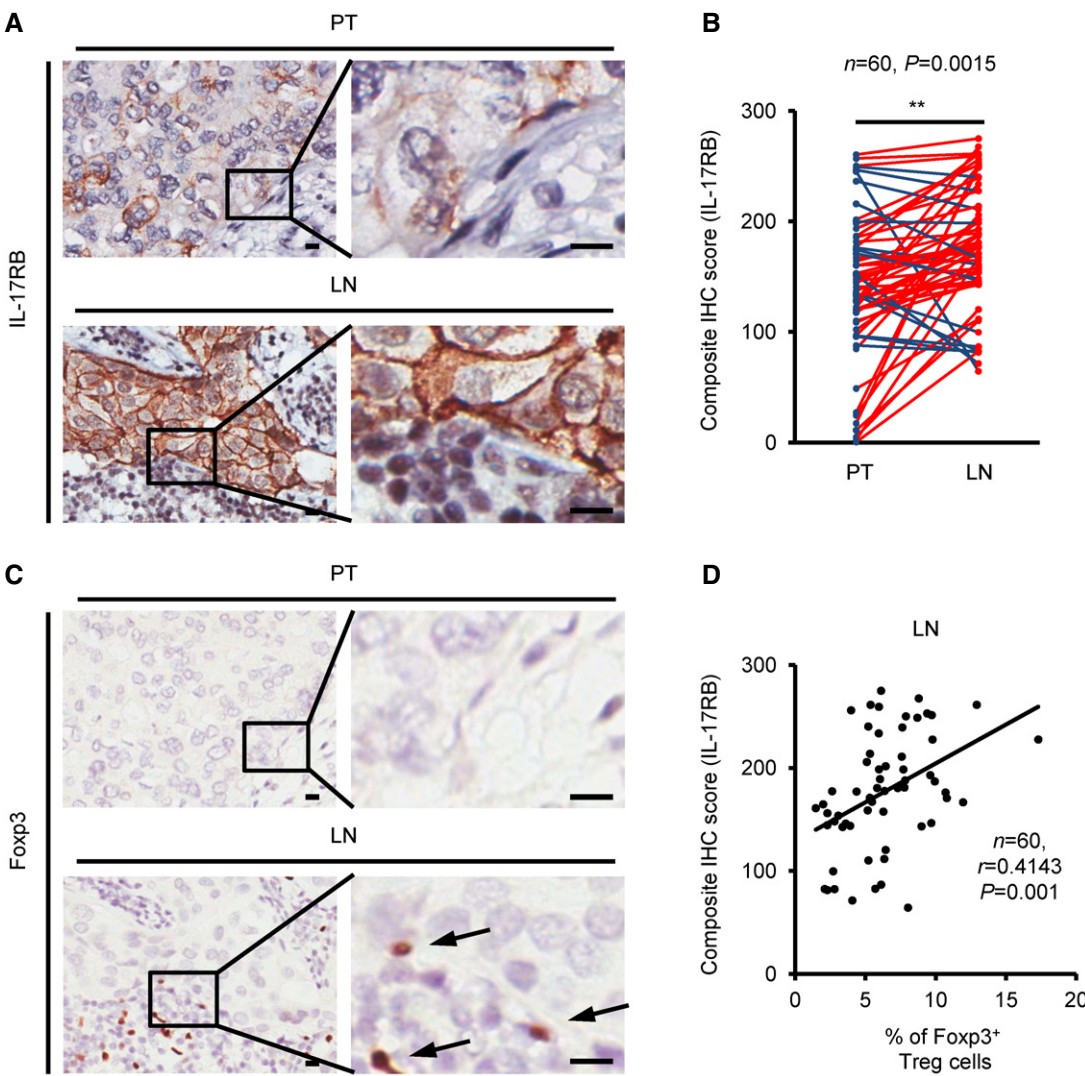

**Figure 7. IL-17RB expression is correlated with the elevated prevalence of regulatory T cells in the tumor-draining lymph nodes of breast cancer patients.**

A–C  (A, C) Representative IHC images of IL-17RB (A) and Foxp3 (C) were taken in breast specimens from primary tumors and their paired LN metastasis specimens. Scale bar: 10 μm. (B) Dot plot showed the results of composite IHC score (staining intensity × percentage of positively stained cell) of IL-17RB from primary tumors (PT; $n = 60$) and their paired LN metastasis (LN; $n = 60$) specimens. Black arrow: Foxp3[+] Tregs cell.

D  The correlation of IL-17RB expression (composite IHC score) and Foxp3[+] Tregs (percentage of total cells) in LN metastasis specimens ($n = 60$). Statistical analysis was performed with the Pearson correlation test.

Data information: In (B), **$P = 0.0015$. Level of significance was determined using Mann–Whitney $U$-test statistic analysis. In (D), $P = 0.001$. Statistical analysis was performed with the Pearson correlation test.

About one-third of breast cancer patients develop distant metastases without invaded cancer cells in LN (Wang *et al*, 2005). This is a strong argument to question about the importance of TDLNs in promoting distant organ metastasis. According to the cohort analysis of 152 breast cancer patients, 73% of node-negative patients are free of distant metastasis 5 years after diagnosis versus 48% of node-positive patients (Rouzier *et al*, 2002). This clinical result suggests a significant role of TDLN in the breast cancer distant organ metastasis, but not absolutely required. Since cancer cell can metastasize to distant organ through either lymphatic or blood circulation (Nathanson *et al*, 2015), this is consistent with our observation that removal of LNs cannot completely abolish metastasis to distant organs (Fig 2D and H).

Micrometastasis in the sentinel LNs of invasive breast cancer is an important indicator for poor survival (Cox *et al*, 2008). Importantly, we found that the GFP[+] 4T1 cells with enhanced malignancy did not cluster and thus may not be detected by IVIS system, but scattered sparsely in the inguinal TDLNs detected by fluorescent microscopy (Fig 1A). Such scattered distribution makes the diagnosis of micrometastasis difficult. Indeed, occult metastasis (metastasis not identified in the initial examination) can be an important prognostic factor for disease recurrence and survival in breast cancer patients (Weaver *et al*, 2011). Re-examination of sentinel LN

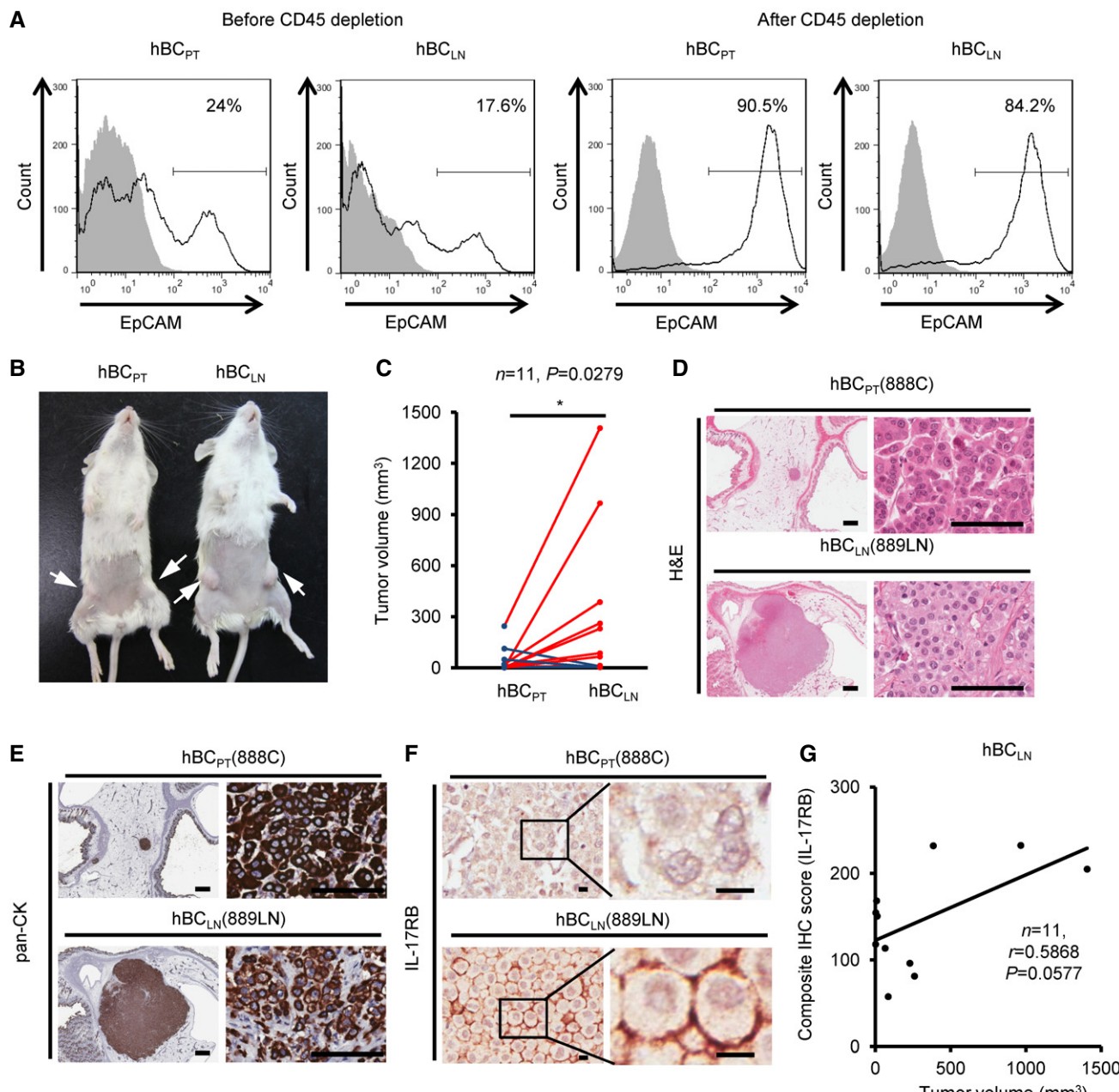

**Figure 8.  Elevated IL-17RB expression in cancer cells derived from tumor-draining lymph nodes in breast cancer patients correlates with aggressive growth nature.**

A    Freshly collected human breast cancer (hBC) specimens (PT, primary tumors; LN, paired LN metastasis from the same breast cancer patient) were digested with collagenase-containing buffer overnight. Depletion of hematopoietic cells by CD45 Dynabeads (Invitrogen) enriched the EpCAM$^+$ cells for following xenografts. Percentages of EpCAM$^+$ cells in human primary breast cancer cells were confirmed by FACS analysis (left two panels: before enrichment; right two panels: after enrichment). All of the histograms were shown that they were EpCAM expressing (black line), compared to isotype control (gray filled).

B    Tumorigenesis assay was determined in NOD/SCID/IL2Rγ$^{null}$ mice orthotopically injected with hBC$_{PT}$ and hBC$_{LN}$ cells. Representative data on day 180 were shown. White arrow: Orthotopic injection site.

C    Dot plot illustrated the tumor volumes of hBC$_{PT}$ and hBC$_{LN}$ cell-injected NOD/SCID/IL2Rγ$^{null}$ mice on day 180 ($n$ = 11). All tumors were confirmed by pan-keratin IHC staining of fat pad sections.

D, E    H&E stainings and IHC stainings of pan-keratin were shown in the representative cases of fat pad sections from (B). Scale bar: 1 mm (left panels), 100 μm (right panels).

F    IHC stainings of IL-17RB were shown in the representative cases of fat pad sections from (B). Scale bar: 10 μm.

G    The correlation of tumor volumes and IL-17RB expression (composite IHC score) in hBC$_{LN}$ cell-injected NOD/SCID/IL2Rγ$^{null}$ mice.

Data information: In (C), *$P$ = 0.0279. Level of significance was determined using Mann–Whitney $U$-test statistic analysis. In (G), statistical analysis was performed with the Pearson correlation test.

biopsies using the National Surgical Adjuvant Breast and Bowel Project (NSABP) B-32 randomized prospective clinical trial showed that patients with occult LN metastases, who previously identified as node-negative in sentinel LN biopsy, had worse overall and disease-free survival as well as distant disease-free interval than patients without occult metastasis (Weaver *et al*, 2011). Thus, thorough examination of metastatic cells in sentinel LN biopsy is needed to improve precise prognosis evaluation.

Tregs in the TDLNs has been thought to suppress anti-tumor immune response (Mansfield *et al*, 2009; Faghih *et al*, 2014). It was reported that tumor-evoked CD25⁺ regulatory B cells (Bregs) induced the expansion of Tregs (Olkhanud *et al*, 2011) to inactivate anti-tumor NK cells and enhance breast cancer lung metastasis in 4T1 mouse model (Olkhanud *et al*, 2009). Thus, depletion of Bregs

by anti-B220 antibody suppresses Tregs expansion and lung metastasis (Olkhanud *et al*, 2011). However, Bregs in TDLNs were not directly promoting tumorigenic and metastatic activities of breast cancer cells (Fig 5C–E). Instead, our results indicated that Tregs in TDLNs directly induced Il-17rb up-regulation of breast cancer cells (Fig 4H). The increased tumorigenic and metastatic activities were predominantly dependent on the induction of Il-17rb (Fig 5B). Thus, it is likely that tumor cells induced Bregs to expand Tregs for at least two potential functions, one is to inactivate NK cells and the other is to up-regulate IL17RB in cancer cells. Since these two groups of Tregs are at different locale, it remains to be investigated whether these two groups of Tregs are identical.

Previously, we reported that the expression of IL-17RB was correlated with poor prognosis in breast cancer patients (Furuta

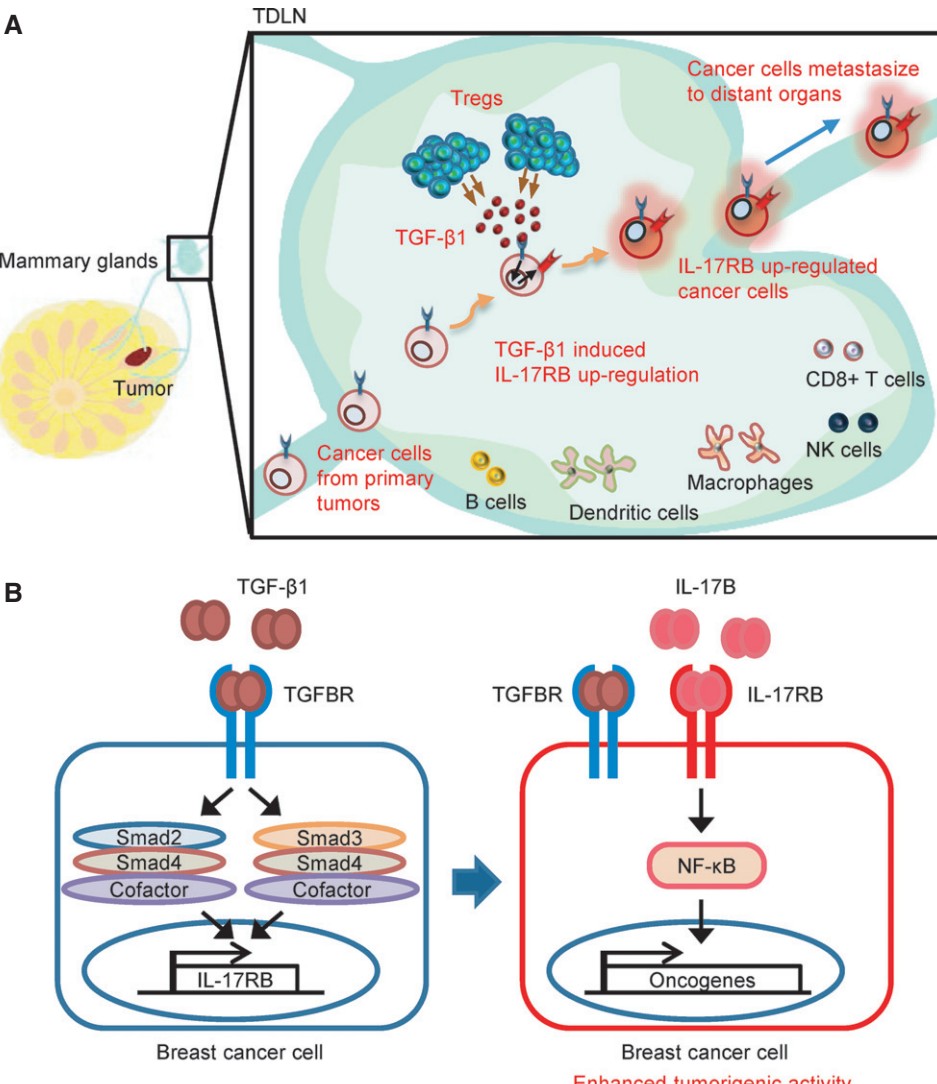

**Figure 9.  Diagram summarizes how breast cancer cells acquire their enhanced malignancy through TDLNs.**

A   TDLNs play a pivotal role in promoting breast cancer cells for distant organ metastasis. Breast cancer cells move to TDLNs where Tregs are elevated and secreted TGF-β1 to up-regulate IL-17RB expression in cancer cells.

B   Expression of IL-17RB in turn activated NF-κB pathway to promote malignancy for distant organ metastasis. Thus, the removal of TDLNs at early time point significantly reduces distant organ metastasis.

*et al*, 2011; Huang *et al*, 2014). Such correlation has also been described in other cancers, including pancreatic and gastric cancer (Wu *et al*, 2015; Bie *et al*, 2016). These studies demonstrated the significance of elevated IL-17B/IL-17RB signaling in promotion of oncogenic activity in cancer cells via either autocrine or paracrine loops. However, less attention was given to the molecular mechanism of how IL-17RB was induced. We showed that the expression of IL-17RB was significantly up-regulated in cancer cells and correlated with the prevalence of Tregs in the LN metastasis (Fig 7D). In the 4T1-allografted mice, we found that Tregs-secreted TGF-β1 derived from TDLNs played a predominant role in the induction of IL-17rb in cancer cells (Fig 6A). The downstream transcription factors of TGF-β1/TGFBR1 signaling, Smad2/3/4, were required for induction of Il-17rb expression after TGF-β1 treatment (Fig 6F–H). Thus, these results showing the regulatory mechanism of IL-17RB expression provided the first mechanistic evidence to highlight the significant role of TDLNs in promoting cancer malignancy.

In addition to the significant roles of *Il-17rb*, other genes up-regulated in the $4T1_{LN}$ cells may also have important roles in promoting cancer malignancy. Genes with a $\log_2$ 1.5-fold up-regulation or down-regulation between $4T1_{PT}$ and $4T1_{LN}$ cells were listed in Tables EV1 and EV2. In a KEGG pathway enrichment analysis of all up-regulated 143 genes, we found that MAPK, PI3K/AKT, and Prolactin signaling pathway were significantly enriched (Appendix Table S1). MAPK (Whyte *et al*, 2009) and PI3K/AKT (Dillon *et al*, 2007) signaling pathways have well-established roles in breast cancer progression. When compared to our previous cDNA microarray raw data of IL-17RB-regulated genes in MDA-MB-361 cells (Huang *et al*, 2014), we found that MAPK signaling pathway was also enriched (Appendix Table S2). Moreover, this group of genes was also increased in $4T1_{LN}$ cells (Appendix Fig S7). However, it remains to be investigated whether this group of genes is directly regulated through the TGF-β1 pathway or IL-17B/IL-17RB pathway or both. Nevertheless, the activation of these genes in $4T1_{LN}$ cells is consistent with the notion of enhanced cancer malignancy.

Cytokines secreted by immune cells in the tumor microenvironment have been found to promote breast cancer progression (Quail & Joyce, 2013). Our findings suggest that TGF-β1 secreted by Tregs in the TDLNs plays a major role in up-regulating Il-17rb in breast cancer cells (Fig 6A). Thus, the higher expression of Il-17rb observed in $4T1_{LN}$ than $4T1_{PT}$ cells (Fig 3A–C) could be due to elevated TGF-β1 in the TDLN and/or up-regulated Tgfbr in $4T1_{LN}$ cells. Since our gene expression profiling showed no difference in *Tgfbr1* and *Tgfbr2* expression between the $4T1_{LN}$ and $4T1_{PT}$ cells (Tables EV1 and EV2), the high level of TGF-β1 in the LN microenvironment should be the dominant cause of Il-17rb induction. The elevated level of secreted TGF-β1 has been shown in the LN metastasis in breast cancer patients (Dalal *et al*, 1993). These observations suggest that the microenvironment of the TDLNs contributes to cancer malignancy via providing high level of TGF-β1 to induce Il-17rb expression in breast cancer cells. However, whether TGF-β1 is the solely factor to induce Il-17rb in breast cancer cells in the TDLNs needs further investigation since other factors such as TNF-α can also up-regulate IL-17RB in primary fibroblast (Kouri *et al*, 2014). Nevertheless, we found that either TGF-β1 neutralization (Fig 6A) or *Tgfbr1* depletion (Fig 6D) in cancer cells significantly abolished Il-17rb up-regulation,

suggesting a critical role of TGF-β1 in up-regulating Il-17rb expression in cancer cells in the TDLNs.

In summary, our study provided clear evidences demonstrating that TDLNs serve as an incubator for cancer cells to enhance their malignancy. In TDLNs, up-regulation of IL-17RB via Tregs-secreted TGF-β1 enhanced malignancy of breast cancer cells as illustrated in Fig 9. This mechanistic finding provides new potential target for blocking distant organ metastasis of breast cancer.

# Materials and Methods

### Cell lines

The mouse mammary carcinoma cell lines 4T1 and EMT6 were obtained from the American Type Culture Collection and maintained in RPMI 1640 or DMEM/F12 medium supplemented with 10% FBS and antibiotics. 4T1 and EMT6 cells were engineered with green fluorescent protein (GFP) and luciferase expression via lentivirus transduction. The pCMV-GFP/luciferase lentivirus (Peng *et al*, 2012) was kindly provided by Dr. Micheal Hsiao (Genomic Research Center, Academia Sinica, Taipei). The cell lines were regularly checked for mycoplasma infections.

### Surgical removal of primary tumors and TDLN

Thirty-eight Balb/c mice were used. For each mouse, $1 \times 10^6$ 4T1 cells mixed with equal volume of Matrigel were injected in the fourth mammary fat pad on the right side of the mouse. Eight days after injection (day 8), the mice were grouped into two groups. The sized-matched tumor-bearing mice were randomly divided into each surgical group. Eighteen mice, under isoflurane anesthesia, were subjected to surgical removal of both the 4T1-derived tumor and the adjacent inguinal LN (Mathieu & Labrecque, 2012). The remaining 20 mice were subjected to tumor removal only. After surgery, metastasis was monitored by bioluminescent imaging on day 14, day 21, day 28, and day 35. Mice with tumor recurrence at the primary sites after tumor removal were excluded according to the bioluminescent imaging on day 14. The investigators were not blinded to the group allocation during experiments and outcome assessment.

### Surgical removal of primary tumors in inguinal LN pre-removed mice

Before 4T1 cells injection, the inguinal LN in the right side of the fourth mammary fat pad was surgically removed from Balb/c mice under isoflurane anesthesia (Mathieu & Labrecque, 2012). The mice in the control group were subjected to a sham surgery. Twelve mice were used in each group. The age- and weight-matched mice were randomly divided into each surgical group. Two weeks after the removal of the inguinal LN, $1 \times 10^6$ 4T1 cells were mixed with equal volume of Matrigel and injected into the same mammary fat pad of both the LN pre-removed and the sham control Balb/c mice. Eight days after injection, the 4T1-derived tumors were surgically removed from mice under isoflurane anesthesia. After surgery, metastasis was monitored by bioluminescent imaging on day 14, day 21, day 28, and day 35. Three inguinal LN pre-removed mice

and seven sham control mice were excluded due to the primary tumor recurrence monitored by bioluminescent imaging on day 14. The investigators were not blinded to the group allocation during experiments and outcome assessment.

## Isolation of mouse breast cancer cells from mice injected with cancer cells and *in vivo* manipulations

To isolate the breast cancer cells from allograft, mice were sacrificed at the indicated time point post-cancer cells injection. To collected $4T1_{PT}$ or $EMT6_{PT}$, primary tumors from both 4th fat pads were excised, minced, and digested with collagenase type I (200 U/ml) and hyaluronidase (50 U/ml) for 16 h in a humidified 37°C incubator supplemented with 5% $CO_2$. Cells were dissociated by 40-μm cell strainers (BD Biosciences, San Jose, CA, USA) and $4T1_{PT}$ cells were maintained in RPMI 1640 medium containing 10% FBS and 60 μM 6-thioguanine (Sigma-Aldrich, St. Louis, MO, USA) and $EMT6_{PT}$ cells were cultured in DMEM/F12 medium supplemented with 10% FBS. To enrich homogenous $4T1_{PT}$ or $EMT6_{PT}$ cells, cells were incubated with APC-conjugated rat anti-mouse CD24 (#101814, Biolegend, San Diego, CA, USA) and PE-conjugated hamster anti-mouse CD29 antibodies (#102208, Biolegend, San Diego, CA, USA) for 30 min at 4°C. The $CD24^+CD29^+$ population (Gao *et al*, 2012) was sorted using a FACS Aria II cell sorter (BD Bioscience, San Jose, CA, USA).

For enrichment of $4T1_{LN}$ cells from tumor-draining LN, inguinal LN as a tumor-draining LN was removed and dissociated by mechanical disruption in the culture dish. All cells were dissociated by 40-μm cell strainers (BD Biosciences, San Jose, CA, USA) and re-suspended in RPMI 1640 medium containing 10% FBS and 60 μM 6-thioguanine (Sigma-Aldrich, St. Louis, MO, USA). After 10–14 days selection/culture, all viable cells were collected and stained with APC-conjugated rat anti-mouse CD24 and PE-conjugated hamster anti-mouse CD29 antibodies for 30 min at 4°C. $CD24^+CD29^+$ $4T1_{LN}$ cells were sorted on a FACS Aria II cell sorter (BD Bioscience, San Jose, CA, USA). For enrichment of $EMT6_{LN}$ cells from tumor-draining LN, total cells from inguinal LN tissues were re-suspended in DMEM/F12 medium supplemented with 10% FBS. After 3–5 days culture, all viable cells were collected and stained with APC-conjugated rat anti-mouse CD24 and PE-conjugated hamster anti-mouse CD29 antibodies for 30 min at 4°C. $CD24^+CD29^+$ $EMT6_{LN}$ cells were sorted by a FACS Aria II cell sorter (BD Bioscience, San Jose, CA, USA). Almost $1 \times 10^5$–$2 \times 10^5$ viable $4T1_{LN}$ cells or $EMT6_{LN}$ cells collected from each LN. The purity of breast cancer cells was greater than 95%.

For *in vivo* tumor growth experiment, 4T1 or EMT6 cells (cell number were indicated in figure legend, respectively) were mixed with equal volume of Matrigel (BD Bioscience, San Jose, CA, USA) and orthotopically injected into both side of the fourth mammary fat pads of BALB/c mice (National Laboratory Animal Breeding and Research Center, Taipei, Taiwan). The age- and weight-matched mice were randomly divided into each experimental group. Tumor growth was assessed morphometrically using electronic calipers, and tumor volumes were calculated according to the formula V ($mm^3$) = L (major axis) × $W^2$ (minor axis)/2 (DuPre *et al*, 2007). Tumor was excised 28 days after initial injection and weighed. For lung colonization assay, Balb/c mice were injected with $2 \times 10^5$ 4T1 cells or $1 \times 10^5$ EMT6 cells through tail vein. Twenty-one days after

injection, lungs were excised and the numbers of tumor nodules were counted. For spontaneous distant organ metastasis assay, 4T1 or EMT6 cells (cell number were indicated in figure legend, respectively) were mixed with equal volume of Matrigel (BD Bioscience, San Jose, CA, USA) and orthotopically injected into both side of the fourth mammary fat pads of NOD/SCID/IL2Rγ$^{null}$ (NSG) mice (kindly provided by Dr. Micheal Hsiao, Genomic Research Center, Academia Sinica, Taipei). The age- and weight-matched mice were randomly divided into each experimental group. For bioluminescent imaging, mice were anaesthetized and intraperitoneally (i.p.) injected with D-luciferin at the indicated time points after cancer cells injection. Mice were imaged in an IVIS 100 chamber within 10 min after D-luciferin injection, and data were recorded using Living Image software (PerkinElmer, Waltham, MA, USA). The *in vivo* depletion of Tregs was performed by i.p. injection of 100 μg rat anti-mouse CD25 antibody (#16-0251, clone PC61, eBioscience, San Jose, CA, USA) at the day 15 and day 18 post-4T1 cells injection. The *in vivo* depletion of $CD4^+$ T cells was performed by i.p. injection of 100 μg rat anti-mouse CD4 antibody (#100416, clone GK1.5, Biolegend, San Diego, CA, USA) at the day 15 and day 18 post-4T1 cells injection.

The investigators were not blinded to the group allocation during experiments and outcome assessment.

## Isolation of human primary breast cancer cell from tumor and LN specimens and mouse xenograft tumor growth assay

For primary human breast cancer cell isolation, tumor specimens from a primary tumor and LN metastasis (NTUH, Taiwan, samples obtained from breast cancer patients with informed consent according to the WMA Declaration of Helsinki and the Department of Health and Human Services Belmont Report) were cut into 2-mm slices and digested in type I collagenase (150 U/ml) and hyaluronidase (50 U/ml) for 16 h in a humidified 37°C incubator supplemented with 5% $CO_2$. After digestion, the tumor tissues were triturated, and the cell suspension was dissociated by 100-μm cell strainers (BD Bioscience, San Jose, CA, USA) and washed with PBS. Cells were maintained in DMEM medium supplemented with 10% FBS.

For xenotransplantation of breast cancer cells derived from fresh human breast cancer specimens, the specimens (PT, primary tumor; LN, paired LN metastasis from the same breast cancer patient) were digested as described above. To enrich the epithelial cell population, we depleted hematopoietic cells using anti-CD45 antibody-coated Dynabeads (Life Technologies, Carlsbad, CA, USA), and isolated breast cancer cells from primary tumor ($hBC_{PT}$) and LN ($hBC_{LN}$). The $hBC_{PT}$ or $hBC_{LN}$ cells ($5 \times 10^5$ cells) were mixed with equal volume of Matrigel and injected into both side of the fourth mammary fat pads of NOD/SCID/IL2Rγ$^{null}$ mice (kindly provided by Dr. Micheal Hsiao, Genomic Research Center, Academia Sinica, Taipei). The age- and weight-matched mice were randomly divided into each experimental group. Tumor volumes were continuously measured, and the final evaluation was made 6 months after initial injection. The investigators were not blinded to the group allocation during experiments and outcome assessment. The significant differences between xenografts of $hBC_{PT}$ and $hBC_{LN}$ cells were determined by Mann–Whitney *U*-test.

## Gene expression using cDNA microarray analysis and quantitative real-time PCR

Total RNAs were extracted from sorted $4T1_{LN}$ and $4T1_{PT}$ breast cancer cells (described above) derived from 4T1 cell-injected mice at the third week with TRIzol® reagent (Life Technologies, Carlsbad, CA, USA), and reversely transcribed with Transcriptor first strand cDNA synthesis kit (Roche, Basel, Switzerland). Microarray analysis was performed using the Mouse Whole Genome OneArray™ (Phalanx Biotech, Palo Alto, CA, USA). The cDNA microarray data have been deposited to the Gene Expression Omnibus (GEO) repository (GEO: GSE86971). To quantify specific gene expression, the quantitative real-time RT–PCR was performed using KAPA SYBR FAST qPCR Kit (KAPA Biosystems, Wilmington, MA, USA) as manufacturer's instruction and analyzed on a Step One Plus Real-Time PCR system (Applied Biosystems, Life Technologies, Carlsbad, CA, USA). Glyceraldehyde 3-phosphate dehydrogenase (*Gapdh*) was used as an internal control for gene expression. All primers were listed in Appendix Table S3.

## Soft-agar colony formation assay

4T1 cells ($5 \times 10^2$ cells) derived from either LNs or primary tumors at different time points after injection were seeded in a layer of 0.35% agar/complete growth medium over a layer of 0.5% agar/complete RPMI1640 medium in a well of a 12-well plate. On day 10–14 after cells seeding, crystal violet-stained colonies were counted.

## shRNA-dependent knockdown and over-expression of Il-17rb by lentiviral transduction

Respective Il-17rb, Gpr56, Scara5, or Tgfbr1 depletion was performed by lentiviral transduction to introduce short hairpin RNAs (shRNA). The pLKOpuro-shIl-17rb (TRCN0000066831, sequence: 5′-cggcaaatggacattctccta-3′), pLKOpuro-shGpr56 (TRCN0000027962, sequence: 5′-gcagaacaccaaagtcaccaa-3′), pLKO-puro-shScara5 (TRCN0000088902, sequence: 5′-gatttggatggac-gatgtgaa-3′), and pLKOpuro-shTgfbr1 (TRCN0000022479, sequence: 5′-gcagagatttatcagactgta-3′) vectors were obtained from National RNAi Core Facility (Taipei, Taiwan). Over-expression of Il-17rb was performed by lentiviral transduction to introduce Il-17rb cDNA. Cloning primers were listed in Appendix Table S3. The Il-17rb cDNA was cloned into pLAS5w.Pbsd-L-tRFP-C vector (National RNAi Core Facility, Taipei, Taiwan). For lentivirus packaging, 293T cells were transfected with pMD.G, pCMVR8.91, and each lentiviral vectors using lipofectamine 2000 (Invitrogen, Life Technologies, Carlsbad, CA, USA) as described previously (Wei *et al*, 2012). Virus-containing culture supernatant was collected and applied for 4T1 cells transduction. The lentivirus-transduced 4T1 cells were selected by puromycin or by FACS sorter, and the expression level of target gene was confirmed by RT–qPCR and Western blotting.

## Genome-editing of mouse *Il-17rb* gene in 4T1 cell

To introduce DNA double-strand break repair-dependent gene deletion or mutation in *Il-17rb* in 4T1 cells, we used RNA-guided endonucleases (RGENs) system (purchased from ToolGen, Geumcheon-gu, Seoul, Korea) to express the Cas9 endonuclease and the guide RNA (gRNA) targeting mouse *Il-17rb* exon 2 (gRNA sequence: 5′-GGAACTCGTCAAGACAAGTG-3′, responding to GRCm38.p2 reference at chr14: 30,006,080–30,006,099). 4T1 cells were transfected with 10 μg of each plasmid using lipofectamine 2000 (Invitrogen, Life Technologies, Carlsbad, CA, USA) following manufacturer's instruction. After 2 days of transfection, we performed limiting dilution to derive single cell clones and measured the Il-17rb expression by Western blotting. To verify the deletion or mutation status of *Il-17rb*, the genomic DNA of *Il-17rb* null clone (the Il-17rb$^{Del}$) was purified and the gRNA-targeted region was amplified by PCR. The PCR products were cloned into the pZBack blunt vector (Tools, New Taipei city, Taiwan), and five of plasmids were sequenced. The primers used for sequencing were described in Appendix Table S3.

## Western blotting

Western blotting was performed as previous reported (Wei *et al*, 2012). Briefly, cells were lysed in RIPA buffer, and protein concentration was determined by Bradford assay (Bio-Rad, Hercules, CA, USA). Equal molarity of protein extracts was loaded and separated in a 10% SDS–PAGE, and transferred to a PVDF membrane. Immunoblot analysis was performed with overnight incubation of 1:50 dilution of mouse anti-GPR56 (#MABN310, Millipore, Temecula, CA, USA), 1:1,000 dilution of mouse anti-IL-17RB (Wu *et al*, 2015), 1:500 dilution of rabbit monoclonal anti-Scara5 (GTX85270, GeneTex, Irvine, CA, USA), 1:2,000 dilution of rabbit anti-Tgfbr1 (GTX102784, GeneTex, Irvine, CA, USA), 1:2,000 dilution of rabbit polyclonal anti-phospho-Thr277-Smad4 (PA5-12685, Thermo Scientific, Rockford, IL, USA), 1:1,000 dilution of rabbit polyclonal anti-Smad4 (GTX112980 GeneTex, Irvine, CA, USA), 1:2,000 dilution of mouse anti-Smad2/3 (#610843, BD Bioscience, San Jose, CA, USA), 1:2,000 dilution of rabbit monoclonal anti-Smad3 (ab40854, abcam, Cambridge, MA, USA), 1:1,000 dilution of rabbit polyclonal anti-Smad6 (#9519, Cell Signaling Technology, Danvers, MA, USA), or 1:1,000 dilution of rabbit polyclonal anti-NF-κB p65 (#3034, Cell Signaling Technology, Danvers, MA, USA) antibody followed by a 1:5,000 dilution of horseradish peroxidase-conjugated anti-mouse or anti-rabbit secondary antibody (GTX221667-01 and GTX221666-01, GeneTex, Irvine, CA, USA). Signals were detected using Immobilon Western Chemiluminescent HRP Substrate (Millipore, Billerica, MA, USA). Mouse anti-GAPDH (1:10,000 dilution, GTX100118, GeneTex, Irvine, CA USA) or Mouse anti-Tubulin (1:5,000 dilution, GTX628802, GeneTex, Irvine, CA USA) antibody was used as loading controls for total cell lysates or cytosolic fraction. Rabbit polyclonal anti-Nuclear Matrix Protein p84 was used as loading control for nuclear fraction (GTX118740, Genetex, Irvine, CA, USA). The intensity of each band was quantified using the ImageJ software (NIH, Bethesda, MD, USA).

## Immunohistochemistry and immunofluorescence

The formalin-fixed, paraffin-embedded tissue sections were treated with antigen retrieval and blocked as preciously described (Huang *et al*, 2014). The slides were incubated with mouse monoclonal anti-cytokeratin (1:500 dilution, GTX75521, GeneTex, Irvine, CA, USA), mouse monoclonal anti-human Foxp3 antibody (1:50

dilution, 14-4777, ebioscience, San Diego, CA, USA), rabbit anti-GFP antibody (1:100 dilution, ADI-SAB-500, Enzo Life Sciences, Farmingdale, NY, USA), or mouse anti-IL-17RB antibody (Wu *et al*, 2015) (1:500 dilution) overnight at 4°C. For immunohistochemistry assay, HRP-conjugated rabbit/mouse polymer (Dako REAL EnVision, Dako, Glostrup, Denmark) and liquid diaminobenzidine tetrahydrochloride plus substrate (DAB chromogen, Dako, Glostrup, Denmark) were used for visualization. All slides were counterstained with hematoxylin, and the images were taken and analyzed using an Aperio Digital Pathology System (Aperio, Vista, CA, USA). For immunofluorescence assay, Alexa Fluor 488-conjugated goat anti-rabbit antibody (1:100 dilution, A11008, Life Technologies, Carlsbad, CA, USA) was used followed by 4′,6-diamidino-2-phenylindole (DAPI) staining. Slides were then mounted with fluorescence mounting medium (Dako, Glostrup, Denmark), and the images were taken by a confocal microscope (Leica TCS SP5 MP).

### FACS analysis

To examine the dynamic change of the subset of immune cell in inguinal LN from mice with 4T1 cells injection, bulk cell populations were collected as described above. Cell sorting was carried out on a FACS Canto cell sorter (BD Bioscience, San Jose, CA, USA). Fluorochrome-conjugated antibodies used for cell surface staining included pacific blue-conjugated rat anti-mouse CD4 (#100428), PE-conjugated rat anti-mouse CD8a (#100708), Alexa Fluor 647-conjugated rat anti-mouse CD3 (#100322), APC-Cy7-conjugated rat anti-mouse B220 (#103224), Alexa Fluor 488-conjugated rat anti-mouse CD11b (#101217), APC-Cy7-conjugated rat anti-mouse CD25 (#102026), Alexa Fluor 488-conjugated rat anti-mouse CD44 (#103016), APC-Cy7-conjugated rat anti-mouse CD49b (#108908), and APC-Cy7-conjugated rat anti-mouse CD62L (#104428) antibodies. All antibodies were purchased from Biolegend (San Diego, CA, USA). For Foxp3, RORγt, and intra-cellular IL-17A staining, cells were fixed and permeabilized with Foxp3 staining buffer set (#560409, BD Bioscience, San Jose, CA, USA) according to the manufacturer's instruction, and then stained with a APC-conjugated rat anti-mouse Foxp3 (#560401, BD Bioscience, San Jose, CA, USA), PE-conjugated rat anti-mouse RORγt (#562607, BD Bioscience, San Jose, CA, USA), and PE-conjugated rat anti-mouse IL-17A (#559502, BD Bioscience, San Jose, CA, USA) antibodies.

To examine the percentage of EpCAM$^+$ cancer cells from human primary breast cancer specimens, viable cells were collected from digested specimens described above. Pacific blue-conjugated mouse anti-human CD45 (#304022, Biolegend, San Diego, CA, USA) and eFluor® 660-conjugated mouse anti-human CD326 antibodies (#50-9326-42, eBioscience, San Diego, CA, USA) were used. Cell sorting analysis was carried out on a FACS Canto cell sorter (BD Bioscience, San Jose, CA, USA).

### Co-culture of breast cancer cells with lymphocytes

For the transwell co-culture experiment, total LN cells were collected from inguinal LN of mice injected with 4T1 cells at the first, second, and third week. To obtain different lymphocyte subsets, the CD4$^+$ lymphocyte, CD8$^+$ lymphocyte, B220$^+$

### The paper explained

#### Problem
Metastasis is the major cause of cancer-related mortality. The presence of cancer cells in tumor-draining lymph nodes (TDLNs) is commonly associated with systemic distant organ metastasis in human breast cancer and is an important prognostic predictor for patient survival. However, whether TDLNs play a significant role in modulating breast cancer cells for distant organ metastasis remains elusive.

#### Results
We found that breast cancer cells isolated from the TDLNs exhibited aggressive phenotypes and removal of TDLNs at early time points significantly reduced distant organ metastasis in a syngeneic breast cancer mouse model. The enhanced malignancy was mainly attributed to the induction of an oncogenic receptor, interleukin-17 receptor B (Il-17rb) in the cancer cells. This induction was initiated by the TGF-β1 secreted from regulatory T cells (Tregs) in the TDLNs. Depletion of Tregs abolished both Il-17rb induction and aggressive phenotypes in breast cancer cells. Importantly, clinical data showed that the expression of IL-17RB in human breast cancer cells in LNs was positively correlated with the prevalence of Tregs. Elevated IL-17RB expression in cancer cells was associated with malignant growth in mouse xenograft assay.

#### Impact
Our study provides the first evidence that the TDLNs serve as an incubator for breast cancer cells to acquire enhanced malignancy. Our findings indicate that IL-17RB can be used as a new diagnostic marker for LN biopsy and a potential target for inhibition of distant organ metastasis in breast cancer.

lymphocyte, CD4$^-$CD8$^-$B220$^-$ lymphocyte, CD4$^+$CD25$^-$ lymphocyte, and CD4$^+$CD25$^+$ lymphocyte subsets were isolated from inguinal LN of mice injected with 4T1 cells at the third week using FACS Aria II cell sorter. The antibodies used for cell surface staining as described above. $3 \times 10^3$ 4T1 cells were seeded in the bottom of a 12-well plate in 1.5 ml of RPMI medium supplemented with 10% FBS. $1 \times 10^7$ of total LN cells or sorted lymphocytes were then seeded on 0.4-mm polyester membrane of a transwell insert (Corning, NY, USA) in 0.5 ml of the same medium. Dishes were incubated for 5 days at a humidified 37 °C incubator supplemented with 5% $CO_2$. For cytokine neutralization experiment, anti-TGF-β1 antibodies (#521704, Biolegend, San Diego, CA, USA, 1 μg/ml), anti-IL-10 antibodies (#16-7101, eBioscience, San Diego, CA, USA, 1 μg/ml), or control IgG was added into a transwell insert.

### Statistical analysis

Except for the clinical specimens and quantification for specific immunoblots and IHC stainings, all data were presented as means ± SD, and two-tailed unpaired *t*-test was used to compare control and treatment groups. Data distribution was assumed to be normal. The variance was similar between the groups that were being statistically compared (by *F*-test). In the analysis of IHC score of patients' specimens and tumor growth assay of patients-derived xenografts, Mann–Whitney *U*-test was used. Asterisk (*) and (**) indicate statistical significance with *P*-value < 0.05 and *P*-value < 0.01, respectively.

## Study approval

Animal studies were approved by the Institutional Animal Care and Use Committee of the Academia Sinica, Taipei, Taiwan (Protocol #14-05-708 and #14-05-709). Human breast cancer specimens and paired LN metastasis specimens were collected from National Taiwan University Hospital. All specimens were encoded to protect patients under protocols approved by the Institutional Review Board of Human Subjects Research Ethics Committee of National Taiwan University, Taipei, Taiwan (IRB no. 200902001R).

## Data availability

The cDNA microarray data have been deposited to the Gene Expression Omnibus (GEO) repository (GEO: GSE86971).

**Expanded View** for this article is available online.

## Acknowledgements

This research was supported by Academia Sinica and Ministry of Science and Technology (MOST 104-0210-01-09-02, MOST 105-0210-01-13-01, MOST 106-0210-01-15-02). Eva YHP Lee was supported by BCRF-35127 grant. S.C.H was supported by an Academia Sinica Postdoctoral Research Fellowship. We thank Drs. Connie Gee and Li-Jung Juan for the critical comments, and Ms Li-Wen Su and Pei-Hsun Chiang for their assistance in this study.

## Author contributions

S-CH, P-CW, WWH-V, C-MH, and W-HL designed the research. S-CH performed all of the experiments. W-HK, Y-MJ, J-YS, C-SH., and K-JC provided essential reagents and clinical specimens. S-CH, WWH-V, P-CW, EY-HPL, and W-HL wrote and completed the manuscript.

## Conflict of interest

The authors declare that they have no conflict of interest.

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
