## [Review Process File · EMBO Molecular Medicine]

TGF- β 1 secreted by Tregs in lymph nodes promotes breast cancer malignancy via up-regulation of IL-17RB

Shih-Chia Huang, Pei-Chi Wei, Wendy W. Hwang-Verslues, Wen-Hung Kuo, Yung-Ming Jeng, Chun-Mei Hu, Jin-Yuh Shew, Chiun-Sheng Huang, King-Jen Chang, Eva Y.-H. P. Lee and Wen-Hwa Lee

Corresponding author: Wen-Hwa Lee, Academia Sinica

Review timeline:

Submission date:	04 August 2016
Editorial Decision:	01 September 2016
Revision received:	18 August 2017
Editorial Decision:	30 August 2017
Revision received:	08 September 2017
Accepted:	12 September 2017

Transaction Report:

Editor: Roberto Buccione

1st Editorial Decision

01 September 2016

Thank you for the submission of your manuscript to EMBO Molecular Medicine. We have now heard back from the three reviewers whom we asked to evaluate your manuscript.

As you will see, while the Reviewers recognize the potential great interest of your work, in aggregate they point to significant and partially overlapping issues that, I am afraid, preclude publication of the manuscript in EMBO Molecular Medicine. I will not dwell into much detail as their comments are detailed. I would like, however, to highlight the main points.

The main fundamental limitations include insufficient support for the main conclusions (e.g. inguinal lymph node metastasis required for lung metastasis), inadequate metastasis modelling in vivo, the use of a single cell line and its variants, and the generally underpowered study (a very important aspect for us at EMBO Molecular Medicine).

After further discussion among the reviewers and with myself, it was unanimously agreed that it is unlikely that these and the other issues can be addressed within a reasonable time frame (3-4 months).

Given these fundamental concerns, and considering that it is our policy to allow only one round of revision, I have no choice but to return the manuscript to you at this stage with the message that we cannot offer to publish it.

I wish to add however that, considered the potential interest of your findings, we would have no

objection to consider a new manuscript on the same topic if at some time in the near future you have obtained data that would considerably strengthen the message of the study by addressing the Reviewers' concerns. Should you decide to follow this route, please make sure you upload a point-by-point rebuttal letter.

I am sorry to have to disappoint you at this stage, and hope the Reviewer comments are useful for your continued work on the topic.

***** Reviewer's comments *****

Referee #1 (Remarks):

In the manuscript entitled, "Tregs secreted TGF β 1 in lymph nodes promote breast cancer malignancy via up-regulation of IL-17RB", Huang et al. report on a communication network between T cells and disseminated cancer cells in the lymph node. The authors show that surgical removal of tumors formed from the transplanted 4T1 mammary cancer cell line together with surgical removal of the inguinal lymph node decreases lung metastasis, as compared with mice whose tumor was removed but whose inguinal lymph node remained intact. The authors show that cancer cells isolated from lymph nodes are more tumorigenic than cancer cells isolated from primary tumors. Lymph node-derived cancer cells are able to form larger numbers of colonies in soft agar and more readily colonize the lung when injected i.v. Comparison of cancer cell from lymph nodes and primary tumors revealed that IL-17RB (as well as other genes) is increased in the lymph nodes. Knocking down IL-17RB reduces colony formation in soft agar, tumor growth in mice and lung colonization after i.v. injection. Through a series of in vitro experiments, the authors identified regulatory T cells as the cell type responsible for IL-17RB up-regulation in 4T1 mammary cancer cells. Depletion of CD25⁺ cells reduces the expression of IL-17RB in cancer cells isolated from lymph nodes. When these cells are re-transplanted or injected i.v., tumor growth and lung colonization is reduced. To regulate the expression of IL-17RB, Tregs secrete TGF β and SMAD proteins are found on the Il-17rb promoter after TGF β treatment of 4T1 cells.

These data follow on from recent papers by the lab of Wen-Hwa Lee, the senior author on the current manuscript, investigating the role of IL-17RB in different cancer types. One publication in *Oncogene* from 2014 (Huang et al. 33, 2968-2977) showed that IL-17RB signaling in human breast cancer cells promotes tumorigenicity and inhibits apoptosis. Interestingly, the authors also showed that IL-17RB expression in human tumor samples correlates with poor prognosis and is associated with HER2⁺ breast cancer, which begs the question why a HER2⁺ model was not used for the purposes of the current study. In 2015, Wen-Hwa Lee's lab reported that IL-17RB overexpression correlates with metastasis in pancreatic cancer (Wu et al. *J Exp Med* 212, 333-49). In similar fashion to previous work in human breast cancer cell lines and the current study using 4T1 mammary cancer cell line, knockdown of IL-17RB in human pancreatic cell lines results in reduced colony formation in soft agar, tumor growth and lung colonization after i.v. injection.

Very little is known about the function of IL-17 family members and their receptors. The current study adds valuable information to the field, uncovering a mechanism by which Treg-derived TGF β regulates IL-17RB expression in metastatic organs. One major caveat to the manuscript is that it contains two concepts that are not necessarily related: 1) lymph node metastasis is required for lung metastasis and 2) TGF β -expressing Tregs induce IL-17RB in 4T1 mammary cancer cells that have metastasized to the lymph node. The data showing that inguinal lymph nodes are required for lung metastasis is remarkable, but these data need further validation. Many conclusions and interpretations are not fully supported by solid evidence. Another general weakness of the manuscript is the lack of details missing from the methodology. I have the following suggestions and comments to improve the manuscript:

1. The data shown in Figure 1 and the author's interpretation that lymph nodes are required for lung metastasis needs further validation. These data suggest that metastatic cancer cells must first travel through the lymphatics, take up residence in the lymph nodes then enter the blood to metastasize to the lung. This concept is not well established and the authors should provide additional experimentation to justify their claim. The use of 5 control mice and 9 experimental mice seems underpowered given the significance of this claim. How many times was the experiment repeated?

Was another model system tested? Is this effect specific to 4T1 mammary cancer cell lines? Does this phenomenon occur in genetically engineered mouse models of breast cancer that better recapitulate tumor progression and metastatic spread? What is the percentage of lymph node-negative breast cancer patients that present with distant metastasis? Is there any evidence in patient cohorts to suggest that lymph node metastasis is required for distant organ metastasis? Moreover, the authors should explain the methodology of these experiments in greater detail. Exactly which inguinal lymph node was removed? Please state which mammary gland the 4T1 cells were injected into and whether the adjacent or opposite inguinal lymph node was removed. The Methods section states "...injected in the 4th pair mammary fat pad..." Does this mean both 4th mammary glands were injected with cells?

2. The use of the term 'xenograft' throughout the manuscript is incorrect. Xenograft refers to the transplantation of material from one species into an entirely different species. Transplantation of 4T1 cells into BALB/c mice is an allograft.

3. Please include details of the 6-thioguanine selection procedure in the Methods section.

4. The figure legend of Figure 1 states that 4T1 cells were selected from lymph nodes and primary tumors by 6-thioguanine for 14 days. How then is it possible that up-regulated expression of IL-17RB in lymph nodes (Figure 2) is maintained for so long, without the influence of Tregs for two weeks?

5. Figures 1d-f show that 4T1 cells isolated from lymph nodes have increased ability to form colonies in soft agar and generate larger tumors in mice. In Figure g-h, the data show that 4T1 cells isolated from lymph nodes form more colonies in the lung when the cells are injected i.v. It is not possible to conclude from this experiment that 4T1LN cells have greater metastatic potential. The data presented by soft agar assays and transplantation into mice provide compelling evidence that 4T1LN cells have greater survival and cell growth properties. Injecting these cells i.v. to induce lung colonization confirms their superior growth potential, not metastasis. Metastasis is the spread of cancer cells from primary tumors to other organs. Injecting cells into the tail vein is not metastasis. As such, there is no evidence to support increased metastasis potential by 4T1LN cells and the authors should remove this conclusion from the manuscript. Figures 1g-h, 2k-l, 4i-j should be reinterpreted.

6. Figure 2: How many mice were analyzed for the cDNA microarray? qPCR? How many replicates were done for Western blots? How many mice were injected in tail vein?

7. Do the "Tregs" isolated for the purposes of Figure 4 suppress T cell proliferation?

8. Anti-CD25 antibodies deplete more than just Tregs and the authors should point this out to the reader. CD25 is expressed by other activated T cell populations including CD8s. Olkhanud et al. (Cancer Research 2011) show that CD25 is also expressed on B cells. The authors should consider depleting all CD4s as well since total CD4s induced the highest expression of IL-17RB in Figure 4d.

9. In relation to Figure 4i-j, the authors state "Since the metastasis ability did not change in 4T1PT collected from the Treg-depleted mice, the increased prevalence of Tregs appeared to play an essential role in promotion of tumor malignancy of 4T1LN in regard to pulmonary metastasis." This interpretation is incorrect since spontaneous metastasis was not tested.

10. In Western blots for IL-17RB expression, there are two bands in Figures 2d, 2f, 2h, 5e but only one band in Figure 3c, 4b, 5d, 5f. Why is this?

11. The authors state that 16 human breast cancer specimens were excluded due to the absence of IL-17RB. In these 16 cases, was there IL-17RB expression in the lymph node in the IL-17RB negative primary tumors?

12. The data in Figure 6d should include more patients, because of the modest p value.

13. I am confused by the data shown in Figure 7. These data show that cancer cells isolated from

lymph nodes grow faster than cancer cells isolated from primary tumors in NSG mice. However, NSG mice do not have Tregs (or T or B or NK cells), so Tregs cannot induce IL-17RB expression in these mice. How do the authors reconcile these conflicting data?

14. In the Discussion, the manuscript states "Blocking cancer cells metastasized to LNs was able to suppress distant organ metastasis in mouse models" and it lists two references: Hirakawa 2007 and Shibata 2008. Neither of these publications provides evidence to support that statement. Hirakawa et al. show that VEGF-C transgenic mice exhibit more lymph node and lung metastasis than controls - this paper does not include blocking experiments. Shibata et al. show that targeting VEGF-A and -C reduces both lymph node and metastasis. Neither of these publications shows that lymph node metastasis is required for lung metastasis.

15. The authors should consult a statistician. Student's t test is not appropriate for the data presented here involving two groups. Mann-Whitney U test should be applied since the data are not normally distributed.

Referee #2 (Remarks):

In this paper, Huang et al examine the role of Treg cells in draining lymph nodes (DLN) on promoting the metastatic potential of breast cancer cells. In particular, the authors propose that TRegs in DLN secrete TGFB1, leading to increased expression of IL17RB in breast cancer cells and consequently a more aggressive metastatic cancer phenotype. This paper extends a similar observation by the same authors (JEM 2015), where they proposed that increased IL17B/IL17RB signaling promotes pancreatic cancer metastasis.

The study correlates the factors influencing the in vitro growth characteristics of mouse mammary carcinoma 4T1 cells with the metastatic behaviour of the same 4T1 cells when transferred to mice in a syngeneic mouse breast tumor model.

Experiments are well performed and clearly presented. In vitro stimulation by isolated TRegs of 4T1 cell expansion is demonstrated to correlate with TGFB1 expression, TReg frequency in vivo in DLN, and the frequency of lung metastases was inhibited by depletion of DLN TRegs. Experiments are presented which support the hypothesis that TRegs in DLN secrete TGFB1, leading to up-regulation of IL17RB in breast cancer cells in this micro-environment and the acquisition of a more aggressive metastatic phenotype.

Overall, this study provides important insight into the possible role of TRegs and the DLN micro-environment in enhancing the metastatic potential of breast cancer cells. While a direct causative link between DLN TRegs and the induction of an increased metastatic phenotype in breast cancer cells as they move through the DLN is not demonstrated, the study provides valuable data from which to examine the role of the DLN micro-environment on tumour cell maturation.

The study might be strengthened by the inclusion of experiments that further explore the functional link between TRegs, TGFB1 expression, IL17RB/IL17B signaling, and tumor metastasis.

Referee #3 (Comments on Novelty/Model System):

As clarified in my response to the authors most conclusions are drawn from 4T1 cell variants and that is the limitation.

Referee #3 (Remarks):

The authors use an established transgenic mouse model of 4T1 cells where they isolated tumor cells from lymph nodes and these displayed a more invasive/aggressive property in breast cancer (BCa) cell growth. The authors provide mechanistic insights of a key role of the IL-17R expression on BCa cells that is triggered by Treg induced TGFBeta expression acting back on the BCa cells. Their results are substantiated by xenograft transplants in immunocompetent syngenic Balb/c recipients and data is also correlated for BCa primary and secondary site metastasis human specimens with Treg analysis. The sole use of Foxp3 staining for Treg quantification on paraffin section is a limited analysis and questionable if state of the art, but the reviewer appreciates the number of paired human

specimens which were pulled into the study which is certainly strength. The last experiment carried out in NSG mice lacks Tregs and is as such inconclusive. Mechanistic insights into the 4T1 cell system are interesting, but the model has questions if really reflecting a physiologic BCa model. Overall, the authors conclude that regulatory T cells in the draining lymph nodes are a key determinant of BCa progression and extravasation through their capacity to secrete TGFbeta to induce IL-17R expression in BCa cells that migrate there. IL-17 stimulates both innate immunity and host defense, where it is e.g. important for colorectal cancer progression. IL-17 plays an active role in inflammatory diseases and to complicate the picture it is also important for autoimmune diseases. IL-17 has a key role in cancer, where multiple receptor chain forms can be expressed and two major cytokine ligands were described. The authors have published a previous manuscript (Huang et al., *Oncogene*, 2014) on the role of IL-17R and NFkappaB activation to be important for BCa cell survival and here they discover Tregs as the main trigger of IL-17R expression, which probably is activated.

Major criticism is given in the following:

1) 4T1 BCa cells are an established model, but this is not a primary BCa or GEMM (numerous genetically engineered mouse models for BCa were made where one GEMM could be evaluated). To proof in a second mouse model setting major findings of the 4T1 model is key for overall conclusion. Results for the 4T1 model are convincing as presented, but a single cell line model has limitations and might have special features that do not closely match a relevant model system for BCa. Two different cell lines or one cell line and one GEMM model based on transgenic or inbred mice seem more justified. Lentiviral integration of reporters were made and the continuous culture of 4T1 cells of unknown passages could result in unwanted superficial BCa cell system which has to be excluded by incorporation of data of a second distinct model. This limitation is the major drawback of the study. The use of a second syngenic Balb/c cell model cell lined called EMT6 is described in the paper foggy. This might be a way to go, but it can nowhere be found in the Figures as clearly marked. In general, a GEMM is superior to another long term culture cell line. The reviewer studied carefully the Figures and legends but results on EMT6 are only speculative from reading Material and Methods section that indicate it was used for the lung invasion assay. This assay is based on tail vein induction, a rather artificial assay that does not reflect physiologic metastatic spread and particular larger tumor cells stick into the blood vasculature of the lung. This has little to do with lymph vessel spreading of BCa cells and can be omitted. To substantiate major conclusions the authors need to question in a second model system in mice in an immunocompetent setting that Tregs reside significantly in the draining lymph nodes and if removed potentially increase survival.

2) IL-17 is largely produced by activated memory T lymphocytes. CD8+ memory effector cells or activated T cells in general express CD25 (B cells or malignant myeloid progenitors can express CD25), which they require for growth and survival. Thus, the use of CD25 antibody injection for Treg ablation has to be described with more care; certainly a genetic deletion of Tregs by knockout of Foxp3 in the T cell lineage e.g. is superior. Moreover, IL-17 can also act on Th17 cells that are also CD4+. Therefore, activated memory effector CD8+ T cells and Th17 cells should be quantified in parallel to Treg quantifications as described in results to Figure 4. Treg or Th17 cells can be grown in vitro similarly after TCR and costimulatory receptor engagement of purified naive CD4+/CD62L+ subsets and both populations expand with stimulation of TGFβ1. Th17 cells require both TGFβ1 and IL-6. To detect both subsets intracellular FACS for Foxp3 and IL-17 or Foxp3 and CD25 could be done. Th17 cells or CD8+ T cells cannot be excluded and could be analyzed in Figure 4.

3) IL-17 signaling can activate different transcription factors (e.g. AP-1, NFkappaB or C/EBP family members) which then might also change their expression levels. The activation of the IL-17R in BCa cells is likely in their 4T1 model system, but it would be important to show it e.g. in the paired 4T1 cells with and without Th17R expression (using the CRISP-Cas9 clone) after pre-stimulation with TGFβ1 to visualize consequences of IL-17 induction on downstream transcription factor action.

Minor:

4) It is questionable if a functional IL-17 response is triggered in the 4T1 cell pairs. Is expression of nuclear factor-kappa B, chemokines CXCL8, CXCL6 and CXCL1 or myeloid cytokines such G-

CSF and GM-CSF, the inflammatory cytokine IL-6 or the adhesion molecule ICAM-1 that were all described to be induced upon IL-17 stimulation distinct in the paired 4T1 cells could add more mechanistic data. A comprehensive microarray would answer that, but focused real time PCR could be done. The authors should provide complete data with significantly up- and down-regulated gene expression in e.g. Suppl. Table format on the microarray mRNA expression analysis and not only show a selected set of five surface marker gene expressions, since that analysis looks narrow or biased. If these data have been generated new it should be described as such. However, if it is from previous work, which was from reading as described unclear, it should be marked as such.

5) The ChIP data in Figure 5h suggest that SMAD proteins are continuously bound at the IL-17rb locus, but SMAD proteins are activated by serine/threonine phosphorylation upon TGFβ stimulation, where they efficiently shuttle to the nucleus. Therefore, the ChIP is unclear and a better analysis with Western blot control on several SMAD family member expressions in BCa cells upon TGFβ stimulation could clarify if indeed pSMAD4 is significantly expressed. Nuclear staining of pSMAD4 could be shown and quantified in respect to another SMAD family member control, protein analysis of 4T1 extracts prior to ChIP by immunoblotting with specific phospho- and total protein antibodies then would allow a better interpretation of transcription factors driving IL-17R expression, which can be driven also by other transcription factors as reasoned above in major point 2.

6) Figure 3e is unclear why not genetic deletion of the clone with CRISP-Cas9 was used as well which could be superior to the shRNA knockdown of 17rb. One reason might be controls, but at least a similar experiment could be performed since the shRNA 4T1 derivatives provide controls and result could be more meaningful.

1st Revision - authors' response

18 August 2017

Detailed point-by-point response to reviewers' comments (bold)

Referee #1 (Remarks):

In the manuscript entitled, "Tregs secreted TGFβ1 in lymph nodes promote breast cancer malignancy via up-regulation of IL-17RB", Huang et al. report on a communication network between T cells and disseminated cancer cells in the lymph node. The authors show that surgical removal of tumors formed from the transplanted 4T1 mammary cancer cell line together with surgical removal of the inguinal lymph node decreases lung metastasis, as compared with mice whose tumor was removed but whose inguinal lymph node remained intact. The authors show that cancer cells isolated from lymph nodes are more tumorigenic than cancer cells isolated from primary tumors. Lymph node-derived cancer cells are able to form larger numbers of colonies in soft agar and more readily colonize the lung when injected i.v. Comparison of cancer cell from lymph nodes and primary tumors revealed that IL-17RB (as well as other genes) is increased in the lymph nodes. Knocking down IL-17RB reduces colony formation in soft agar, tumor growth in mice and lung colonization after i.v. injection. Through a series of in vitro experiments, the authors identified regulatory T cells as the cell type responsible for IL-17RB up-regulation in 4T1 mammary cancer cells. Depletion of CD25+ cells reduces the expression of IL-17RB in cancer cells isolated from lymph nodes. When these cells are re-transplanted or injected i.v., tumor growth and lung colonization is reduced. To regulate the expression of IL-17RB, Tregs secrete TGFβ and SMAD proteins are found on the IL-17rb promoter after TGFβ treatment of 4T1 cells.

These data follow on from recent papers by the lab of Wen-Hwa Lee, the senior author on the current manuscript, investigating the role of IL-17RB in different cancer types. One publication in Oncogene from 2014 (Huang et al. 33, 2968-2977) showed that IL-17RB signaling in human breast cancer cells promotes tumorigenicity and inhibits apoptosis. Interestingly, the authors also showed that IL-17RB expression in human tumor samples correlates with poor prognosis and is associated with HER2+ breast cancer, which begs the question why a HER2+ model was not used for the purposes of the current study. In 2015, Wen-Hwa Lee's lab reported that IL-17RB overexpression correlates with metastasis in pancreatic cancer (Wu et al. J Exp Med 212, 333-49). In similar fashion to previous work in

human breast cancer cell lines and the current study using 4T1 mammary cancer cell line, knockdown of IL-17RB in human pancreatic cell lines results in reduced colony formation in soft agar, tumor growth and lung colonization after i.v. injection.

Very little is known about the function of IL-17 family members and their receptors. The current study adds valuable information to the field, uncovering a mechanism by which Treg-derived TGF β regulates IL-17RB expression in metastatic organs.

Response: We appreciate the reviewer's encouragement.

One major caveat to the manuscript is that it contains two concepts that are not necessarily related: 1) lymph node metastasis is required for lung metastasis and 2) TGF β -expressing Tregs induce IL-17RB in 4T1 mammary cancer cells that have metastasized to the lymph node. The data showing that inguinal lymph nodes are required for lung metastasis is remarkable, but these data need further validation.

Response: We apologize for not clarifying the point that leads to the impression of the reviewer. In our paper, we like to emphasize that (1) the tumor cells metastasized to lymph node (LN) gain much strong malignancy than the primary cells. Those cells have stronger metastatic ability than the primary cells. LNs play an important role, but are not required for metastasis because cancers have different routes to move out from the primary site. The second point of the paper is to elucidate (2) how LN microenvironment reprograms the cancer cells to gain the strong malignancy activity. With this clarification, we believe that the data provided are very solid and conclusive as described below.

I have the following suggestions and comments to improve the manuscript:

Specific comments:

1. The data shown in Figure 1 and the author's interpretation that lymph nodes are required for lung metastasis needs further validation. These data suggest that metastatic cancer cells must first travel through the lymphatics, take up residence in the lymph nodes then enter the blood to metastasize to the lung. This concept is not well established and the authors should provide additional experimentation to justify their claim. The use of 5 control mice and 9 experimental mice seems underpowered given the significance of this claim. How many times was the experiment repeated?

Response: We apologized for the confusion. Tumor-draining lymph nodes (TDLNs) are important for enhancing breast cancer cells metastasis ability, but are not required for breast cancer metastasis to the lung. We employed additional experiments described below to strengthen this observation.

The original Figure 1a-1c and Supplementary Figure 2 were pooled data obtained from three independent experiments. To increase the sample size and to show that these results are reproducible, we performed another experiment using 20 control mice (surgical removal of a primary tumor) and 18 experimental mice (surgical removal of both primary tumor and inguinal LN). One week after tumor resection, 10 control mice and 5 experimental mice had tumor growth at the primary sites (Figure R1) and were excluded from the experiment because a few residual 4T1 tumor cells may not be completely removed after surgery of the primary tumors. At four weeks post tumor resection, we took IVIS images and observed a lower percentage (3 out of 13 mice) of distant organ metastasis in experimental mice comparing with the control mice (5 out of 10 mice). Although not all of the control mice (5 out of 10 mice) showed distant organ metastasis at this time (Figure R2), we still found TDLN removal significantly decrease the percentage of distant organ metastasis in 4T1 mouse model. These data is now presented in the Figure 2A-2E and the Supplementary Figure 3 of the revised manuscript.

Original Figure 1a-1c: Removal of tumor-draining lymph node reduces distant organ metastasis of breast cancer cells.

Original Supplementary Figure 2: IVIS images from control mice and experimental mice.

Figure R1: Ten control mice (surgical removal of a primary tumor) and five experimental mice (surgical removal of both primary tumor and inguinal LN) were excluded due to tumor growth at the primary sites one week after tumor resection. The excluded mice are labeled in red. These results are now presented in the revised manuscript.

Figure R2: Removal of tumor-draining lymph node reduces distant organ metastasis of breast cancer cells in 4T1 mouse model. These results are now presented in the revised manuscript.

Was another model system tested? Is this effect specific to 4T1 mammary cancer cell lines?

Response: In addition to 4T1 cells, we used another mouse breast cancer cell line, EMT6 cells, to repeat this experiment. Unfortunately, we did not observe any distant organ metastasis four weeks after tumor resection in either control or experimental mice. We continued to monitor these mice until week 9, but still did not observe distant organ metastasis in either group (Figure R3). This result may be explained by the lower metastatic ability of EMT6 cells as reported in a recent study (Ouzounova et al, 2017). To obtain a sub-clone of EMT6 cells with higher metastatic activity, we injected EMT6 cells in one of the fourth mammary fat pads, waited for 10 weeks and isolated metastasized EMT6 cells from the lung for culture. We then used this lung meta-derived EMT6 sub-clone (LM-EMT6) to perform the experiment described in Figure R1. As shown in Figure R4, we quickly observed 3 out of 9 experimental mice with distant organ metastasis two weeks after tumor resection. However, the recurrence of primary tumor was very high, 8 out of 9, in the control mice group, this made the result less conclusive though showed a positive trend (Figure R4).

Figure R3: Distant organ metastasis did not occur after EMT6-derived tumors were removed.

Figure R4: The high percentage of primary tumor recurrence after tumor resection in the EMT6 mouse model when using lung meta-derived EMT6 sub-clone.

To further demonstrate the role of TDLN in distant organ metastasis of breast cancer, we designed another experiment with a different lymph node surgical removal schedule. For each experimental mouse, the inguinal lymph node in the fourth mammary fat pad on the right side of the mouse was removed two weeks before injecting the 4T1 cells in the same fat pad. A sham surgery was performed on the control mice at the same schedule. Eight days after 4T1 injection, we removed the primary tumors from these mice and then monitored distant organ metastasis by IVIS imaging. The mice with tumor recurrence at the primary sites after tumor removal were excluded (Figure R5). We observed a relatively lower percentage (1 out of 9 mice) of distant organ metastasis in the inguinal LN pre-removal mice (experimental mice) compared to the sham surgery control mice (3 out of 5 control mice) (Figure R6). This data is now presented in the Figure 2F-2I and Supplementary Figure 4 of the revised manuscript. Together, the results in Figure R2 and R6 both illustrated that the TDLN plays a significant role in distant organ metastasis of breast cancer.

Figure R5: Seven control mice (surgical removal of a primary tumor in sham surgery mice) and three experimental mice (surgical removal of a primary tumor in lymph node-removed mice) are excluded due to the primary tumor recurrence at week 2 and labeled in red. These results are now presented in the revised manuscript.

Figure R6: Pre-removal of inguinal lymph node reduces distant organ metastasis of breast cancer cells in 4T1 mouse model. These results are now presented in the revised manuscript.

Does this phenomenon occur in genetically engineered mouse models of breast cancer that better recapitulate tumor progression and metastatic spread?

Response: We agree with the reviewer that a genetically engineered mouse model may be better than syngeneic allograft mouse model. However, to generate such a model and use such model for this question may not be a straightforward task.

What is the percentage of lymph node-negative breast cancer patients that present with distant metastasis? Is there any evidence in patient cohorts to suggest that lymph node metastasis is required for distant organ metastasis?

Response: LN status is an established prognostic marker of breast cancer metastasis for clinical use, and the presence of LN metastasis increases the risk of distant metastasis development (Weigelt et al, 2005). Although one-third of breast cancer patients develop distant metastases without invaded cancer cells in LN (Wang et al, 2005), a several patient cohort studies provide evidence to demonstrate a significant association between LN metastasis and distant organ metastasis, and survival (Ran et al, 2010). In a cohort analysis of 152 breast cancer patients, 73% of node-negative patients are free of distant metastasis five years after diagnosis versus 48% of node-positive patients (Rouzier et al, 2002). These clinical results suggest a significant role of TDLNs in the breast cancer distant organ metastasis, but not absolutely required.

Moreover, the authors should explain the methodology of these experiments in greater detail. Exactly which inguinal lymph node was removed? Please state which mammary gland the 4T1

cells were injected into and whether the adjacent or opposite inguinal lymph node was removed. The Methods section states "...injected in the 4th pair mammary fat pad..." Does this mean both 4th mammary glands were injected with cells?

Response: We apologize that our description in the original manuscript was unclear. A more detailed methodology is described below and presented in the revised manuscript:

Surgical removal of primary tumors and TDLN: Thirty eight Balb/c mice were used. For each mouse, 106 4T1 cells mixed with equal volume of Matrigel were injected in the 4th mammary fat-pad on the right side of the mouse. Eight days after injection (day8), the mice were grouped into two groups. Eighteen mice, under isoflurane anesthesia, were subjected to surgical removal of both the 4T1-derived tumor and the adjacent inguinal LN. The remaining 20 mice were subjected to tumor removal only. After surgery, metastasis was monitored by bioluminescent imaging on day14, day21, day28, and day35. Mice with tumor recurrence at the primary sites after tumor removal were excluded according to the bioluminescent imaging on day14.

Surgical removal of primary tumors in inguinal LN pre-removed mice: Before 4T1 cells injection, the inguinal LN in the right side of the 4th mammary fat pad was surgically removed from Balb/c mice under isoflurane anesthesia. The mice in the control group were subjected to a sham surgery. Twelve mice were used in each group. Two weeks after the removal of the inguinal LN, 106 4T1 cells were mixed with equal volume of Matrigel and injected into the same mammary fat-pad of both the LN pre-removed and the sham control Balb/c mice. Eight days after injection, the 4T1-derived tumors were surgically removed from mice under isoflurane anesthesia. After surgery, metastasis was monitored by bioluminescent imaging on day14, day21, day28, and day35. Three inguinal LN pre-removed mice and seven sham control mice were excluded due to the primary tumor recurrence monitored by bioluminescent imaging on day14.

2. The use of the term 'xenograft' throughout the manuscript is incorrect. Xenograft refers to the transplantation of material from one species into an entirely different species. Transplantation of 4T1 cells into BALB/c mice is an allograft.

Response: We thank the reviewer's correction. The term has been changed to allograft in the revised manuscript.

3. Please include details of the 6-thioguanine selection procedure in the Methods section.

Response: The 6-thioguanine selection process is described in the methods section in the revised manuscript:

Isolation of mouse breast cancer cells from mice injected with cancer cells and *in vivo* manipulations: To isolate the breast cancer cells from allograft, mice were sacrificed at the indicated time point post cancer cells injection. To collect 4T1PT or EMT6PT, primary tumors from both 4th fat pads were excised, minced and digested with collagenase type I (200 U/ml) and hyaluronidase (50 U/ml) for 16 hr in a humidified 37 °C incubator supplemented with 5% CO₂. Cells were dissociated by 40 µm cell strainers (BD Biosciences, San Jose, CA, USA) and 4T1PT cells were maintained in RPMI 1640 containing 60 µM 6-thioguanine (Sigma-Aldrich, St. Louis, MO, USA) and EMT6PT cells were cultured in DMEM/F12 medium supplemented with 10 % FBS. To enrich homogenous 4T1PT or EMT6PT cells, cells were incubated with APC-conjugated rat anti-mouse CD24 (#101814, Biolegend, San Diego, CA, USA) and PE-conjugated hamster anti-mouse CD29 antibodies (#102208, Biolegend, San Diego, CA, USA) for 30 mins at 4°C. The CD24+CD29+ population (Gao et al, 2012) was sorted using a FACS Aria II cell sorter (BD Bioscience, San Jose, CA, USA).

For enrichment of 4T1LN cells from tumor-draining LN, inguinal LN as a tumor-draining LN was removed and dissociated by mechanical disruption in the culture dish. All cells were dissociated by 40 µm cell strainers (BD Biosciences, San Jose, CA, USA) and re-suspended in RPMI 1640 medium containing 60 µM 6-thioguanine (Sigma-Aldrich, St. Louis, MO, USA) supplemented with 10 % FBS. After 10-14 days selection/culture, all viable cells were collected and stained with APC-conjugated rat anti-mouse CD24 and PE-conjugated hamster anti-mouse CD29 antibodies for 30 mins at 4°C. CD24+CD29+ 4T1LN cells were sorted on a FACS Aria II cell sorter (BD Bioscience, San Jose, CA, USA). For enrichment of EMT6LN cells from tumor-draining LN, total cells from

inguinal LN tissues were re-suspended in DMEM/F12 medium supplemented with 10 % FBS. After 3-5 days culture, all viable cells were collected and stained with APC-conjugated rat anti-mouse CD24 and PE-conjugated hamster anti-mouse CD29 antibodies for 30 mins at 4°C. CD24+CD29+ EMT6LN cells were sorted by a FACS Aria II cell sorter (BD Bioscience, San Jose, CA, USA). Almost 1×10^5 - 2×10^5 viable 4T1LN cells or EMT6LN cells collected from each LN. The purity of breast cancer cells was greater than 95%.

4. The figure legend of Figure 1 states that 4T1 cells were selected from lymph nodes and primary tumors by 6-thioguanine for 14 days. How then is it possible that up-regulated expression of IL-17RB in lymph nodes (Figure 2) is maintained for so long, without the influence of Tregs for two weeks?

Response: We harvested all cells from LN and put them into culture dish with complete RPMI1640 medium with 10% FBS and all supplements. This medium is also used for mouse lymphocyte cultures and can keep the viability of lymphocyte for a week. The addition of 6-TG in culture dish was used for inhibiting the activated lymphoblast proliferation during the 10-14 days culture. We think that Tregs may survive in the first week, and produce TGF- β 1 to maintain the expression of IL-17rb in 4T1 cells. This statement can be supported by our results in the original Fig. 3c, 4a, 4b, and 4d. IL-17rb expression was significantly induced in 4T1 cells after five-day transwell co-culture with lymphocytes isolated from TDLNs (Original Fig. 3c). After lymphocytes died out, 4T1 cells may still keep IL-17rb expression through epigenetic regulation similar to what we observed in the co-culture of human breast cancer cell line with normal tissue-associated fibroblasts previously (Tyan et al, 2012; Tyan et al, 2011). However, when the epigenetic modifications occurred during the selection culture of 4T1 cells needs further investigation.

c

Original Figure 3c: The IL-17rb expression in 4T1 cells is increased when co-cultured with cells from tumor-draining LNs.

a

b

Original Figure 4a and 4b: The significant induction of IL-17rb is observed only when 4T1 cells are co-cultured with CD4⁺ T cell subset.

Original Figure 4d: The CD4⁺CD25⁺ T cells, but not CD4⁺CD25⁻ T cells, are able to induce *Il-17rb* expression of 4T1 cells in the transwell co-culture assay.

5. Figures 1d-f show that 4T1 cells isolated from lymph nodes have increased ability to form colonies in soft agar and generate larger tumors in mice. In Figure g-h, the data show that 4T1 cells isolated from lymph nodes form more colonies in the lung when the cells are injected i.v. It is not possible to conclude from this experiment that 4T1LN cells have greater metastatic potential. The data presented by soft agar assays and transplantation into mice provide compelling evidence that 4T1LN cells have greater survival and cell growth properties. Injecting these cells i.v. to induce lung colonization confirms their superior growth potential, not metastasis. Metastasis is the spread of cancer cells from primary tumors to other organs. Injecting cells into the tail vein is not metastasis. As such, there is no evidence to support increased metastasis potential by 4T1LN cells and the authors should remove this conclusion from the manuscript. Figures 1g-h, 2k-l, 4i-j should be reinterpreted.

Response: We thank the reviewer's comment. We corrected our interpretation of the results shown in the original Fig. 1g-h, Fig. 2k-l, and Fig. 4i-j. These results were now shown in the Figure 1F-1G, and Figure 3K-L of the revised manuscript.

To further demonstrate that 4T1LN cells have greater metastatic potential than 4T1PT cells, we isolated and injected 4T1LN, 4T1PT, EMT6LN and EMT6PT cells into both side of the 4th mammary fat pads of immuno-deficient NSG mice. We monitored the distant organ metastasis by IVIS imaging. As shown in Figure R7, a higher percentage of distant organ metastasis in 4T1LN- or EMT6LN-injected NSG mice compared to 4T1PT- or EMT6PT-injected NSG mice was observed. Thus, this new results provide evidence that 4T1LN cells have greater metastatic potential than 4T1PT cells. This new data is presented in the Figure 1H-1I of the revised manuscript.

Figure R7: 4T1LN cells (left panel) and EMT6LN (right panel) cells showed greater metastatic potential than 4T1PT and EMT6PT cells. This data is now presented in the revised manuscript.

6. Figure 2: How many mice were analyzed for the cDNA microarray? qPCR? How many replicates were done for Western blots? How many mice were injected in tail vein?

Response: We used at least five 4T1 tumor-bearing mice to obtain enough FACS-sorted 4T1LN cells or 4T1PT cells for mRNA extraction. cDNA microarray was performed using the Mouse Whole Genome OneArray™ (Phalanx Biotech, Palo Alto, CA, USA). The cDNA microarray data is deposited to the Gene Expression Omnibus (GEO) repository (GEO: GSE86971).

For each qPCR and Western blotting experiment, we used pooled 4T1LN cells or 4T1PT cells from two to three mice to collect enough material. All data were verified in at least two independent experiments. For the tail-vein injection experiment, three mice were used in each group in each experiment.

7. Do the "Tregs" isolated for the purposes of Figure 4 suppress T cell proliferation?

Response: To answer this question, we isolated Tregs from the inguinal LNs of the WT control mice and the 4T1-tumor bearing mice by FACS sorting (CD4+CD25+GITR+cells). The sorted Tregs were then co-cultured with CFSE-labeled CD3+ T cells from WT mice to examine their suppressive function. As shown in Figure R8, Tregs isolated from 4T1-tumor bearing mice remain exhibiting normal T cell suppressive activity at week3.

Figure R8: Tregs isolated from inguinal LNs in WT control mice (WT Tregs) and 4T1-tumor bearing mice (4T1 Tregs) remain exhibiting T cell suppressive activity at week3.

8. Anti-CD25 antibodies deplete more than just Tregs and the authors should point this out to the reader. CD25 is expressed by other activated T cell populations including CD8s. Olkhanud et al. (Cancer Research 2011) show that CD25 is also expressed on B cells. The authors should consider depleting all CD4s as well since total CD4s induced the highest expression of IL-17RB in Figure 4d.

Response: We apologize for missing this information in the original manuscript. We added the information that the anti-CD25 antibodies (Abs) deplete more than just Tregs in the revised manuscript.

We appreciated the reviewer's suggestion and repeated the experiments using anti-CD4 Abs or anti-CD25 Abs to deplete total CD4⁺ T cells or CD25⁺ cells respectively in 4T1 tumor-bearing mice. The 4T1LN were then isolated from the total CD4⁺ T cells-depleted mice, CD25⁺ cells-depleted mice or control IgG-treated mice for further analyses. We found that both anti-CD4 Abs and anti-CD25 Abs injection can significantly decrease the Treg subset in the TDLN of the 4T1 tumor-bearing mice (Figure R9a). The expression of *Il-17rb* in the 4T1LN cells collected from either CD4⁺ T cells-depleted mice or CD25⁺ cells-depleted mice was significantly abolished (Figure R9b). Moreover, both 4T1LN derived tumor growth and distant organ metastasis (IVIS image was taken at day 21) in the CD4⁺ T cells-depleted mice and the CD25⁺ cells-depleted mice (Rebuttal Figure 9c-e) were significantly reduced. These results are now presented in Figure 5 of the revised manuscript.

Figure R9: Depleting total CD4⁺ T cells or CD25⁺ cells in the tumor draining LN (a) abolished Il-17rb induction (b), tumor growth ability (c and d), and metastatic activity of 4T1LN cells (e). This data is now presented in the revised manuscript.

9. In relation to Figure 4i-j, the authors state "Since the metastasis ability did not change in 4T1PT collected from the Treg-depleted mice, the increased prevalence of Tregs appeared to play an essential role in promotion of tumor malignancy of 4T1LN in regard to pulmonary metastasis." This interpretation is incorrect since spontaneous metastasis was not tested.

Response: We now make the conclusion based on the new results described in Figure R9 (comment 8). Figure R9e showed that the spontaneous distant organ metastasis of 4T1LN from the CD4⁺ T cells-depleted mice or CD25⁺ cells-depleted mice was significantly reduced. However, this inhibition of spontaneous metastasis ability was not found in 4T1PT isolated from CD4⁺ T cells-

depleted or CD25⁺ cells-depleted mice when compared with 4T1PT isolated from control IgG-treated mice.

10. In Western blots for IL-17RB expression, there are two bands in Figures 2d, 2f, 2h, 5e but only one band in Figure 3c, 4b, 5d, 5f. Why is this?

Response: These inconsistent patterns were due to the cropped scan of Western blots. The MW of Il-17rb protein in mouse cell is around 70kDa, and a non-specific band was routinely found in a region between 55 to 70kDa. To clarify it, we have provided our uncropped scan of all Western blots in the supplemental information in the revised manuscript and below (Figure R10).

Figure R10a: Uncropped scans of Immunoblots shown in Original Figure 2d, 2f, 2h, 3c, and 4b.

Figure R10b: Uncropped scans of Immunoblots shown in Original Figure 5d, 5e, and 5f.

11. The authors state that 16 human breast cancer specimens were excluded due to the absence of IL-17RB. In these 16 cases, was there IL-17RB expression in the lymph node in the IL-17RB negative primary tumors?

Response: None of these 16 cases had detectable IL-17RB expression in the lymph nodes and the primary tumors using our assay system.

12. The data in Figure 6d should include more patients, because of the modest p value.

Response: We have collected additional 26 new cases with detectable IL-17RB in either the lymph nodes or primary tumors. As shown in Figure R11, a significant p-value ($p=0.001$, Pearson $r=0.4143$) was found after addition of these new cases. These results are now presented in Figure 7 of the revised manuscript.

Figure R11: IL-17RB expression is correlated with the elevated prevalence of regulatory T cells in the tumor-draining lymph nodes of breast cancer patients. Representative IHC images of IL-17RB (a) and Foxp3 (c) were taken in breast specimens from primary tumors (PT; $n=60$) and their paired LN metastasis specimens (LN; $n=60$). Scale bar : 10 μ m. (b) Dot plot showed the results of composite IHC score (staining intensity \cdot percentage of positively stained cell) of IL-17RB from primary tumors and their paired LN metastasis specimens. (d) The correlation of IL-17RB expression (composite IHC score) and Foxp3+ Treg cells (percentage of total cells) in LN metastasis specimens.

13. I am confused by the data shown in Figure 7. These data show that cancer cells isolated from lymph nodes grow faster than cancer cells isolated from primary tumors in NSG mice. However, NSG mice do not have Tregs (or T or B or NK cells), so Tregs cannot induce IL-17RB expression in these mice. How do the authors reconcile these conflicting data?

Response: We apologize if our description in the original manuscript was confusing. In figure 7, we injected the primary human breast cancer cells which freshly isolated from patient's specimens. hBCPT cells came from the primary tumor site, and hBCLN cells came from the tumor adjacent LN metastasis. After isolation of hBCPT cells and hBCLN cells from the same patient, we mixed cells with Matrigel and injected it into immuno-deficient NSG mice respectively to analyze their tumorigenic activity.

These cancer cells were already interacted with their tumor microenvironment either at the primary site or TDLN in the patients for a long time. It is possible that the hBCLN cells in the TDLN have been "educated" long enough that the expression of IL-17RB in the hBCLN has become constantly up-regulated by epigenetic machinery. Thus, when evaluating the tumorigenic activities of the hBCPT and the hBCLN cells in mice without Tregs, hBCLN still grew faster. We described this experimental procedure in the methods section in the manuscript and below:

Isolation of primary human breast cancer cell from tumor and LN specimens and mouse xenograft tumor growth assay: For primary human breast cancer cell isolation, tumor specimens from a

primary tumor and LN metastasis were cut into 2-mm slices and digested in type I collagenase (150 U/ml) and hyaluronidase (50 U/ml) for 16 hr in a humidified 37°C incubator supplemented with 5% CO₂. After digestion, the tumor tissues were triturated, and the cell suspension was dissociated by 100 µm cell strainers (BD Bioscience, San Jose, CA, USA) and washed with PBS. Cells were maintained in DMEM medium supplemented with 10% FBS.

For xenotransplantation of breast cancer cells derived from fresh human breast cancer specimens, the specimens (PT, primary tumor; LN, paired LN metastasis from the same breast cancer patient) were digested as described above. To enrich the epithelial cell population, we depleted hematopoietic cells using anti-CD45 antibody coated Dyna beads (Life Technologies, Carlsbad, CA, USA), and isolated breast cancer cells from primary tumor (hBCPT) and LN (hBCLN). The hBCPT or hBCLN cells (5x10⁵ cells) were mixed with equal volume of Matrigel and injected into the mammary fat-pad of NOD/SCID/IL2R γ null mice (kindly provided by Dr. Micheal Hsiao, Genomic Research Center, Academia Sinica, Taipei). Tumor volumes were continuously measured, and the final evaluation was made six months after initial injection. The significant differences between xenografts of hBCPT and hBCLN cells were determined by Mann-Whitney U test.

14. In the Discussion, the manuscript states "Blocking cancer cells metastasized to LNs was able to suppress distant organ metastasis in mouse models" and it lists two references: Hirakawa 2007 and Shibata 2008. Neither of these publications provides evidence to support that statement. Hirakawa et al. show that VEGF-C transgenic mice exhibit more lymph node and lung metastasis than controls - this paper does not include blocking experiments. Shibata et al. show that targeting VEGF-A and -C reduces both lymph node and metastasis. Neither of these publications shows that lymph node metastasis is required for lung metastasis.

Response: We apologized for the inadequate explanation of these papers. We have revised our discussion as below.

"There has been a long-standing debate over the role of LNs in promoting cancer malignancy. Some clinical evidence has suggested that metastasis to the LNs in breast cancer patients is strongly associated with distant organ metastasis, poor disease-free survival and shorter overall survival (Ran et al, 2010; Rouzier et al, 2002). Although breast cancer cells-induced lymphangiogenesis in TDLNs is important to distant organ metastasis in mouse model, there is no direct evidence to demonstrate that breast cancer cells metastasized to LNs was required for further distant organ metastasis (Hirakawa et al, 2007; Shibata et al, 2008). Several clinical trials showed no survival benefits for patients underwent lymphadenectomy (Gervasoni et al, 2007). Thus, whether TDLNs involved in the progression of systemic metastasis remain controversial (Pereira et al, 2015; Ran et al, 2010). Using a syngeneic mouse model, we observed that breast tumor cells derived from TDLN gained higher malignancy compared with that from the primary site and removal of TDLNs significantly reduced distant metastasis. Further investigation showed that cancer cells in the TDLNs enhanced their malignancy by up-regulating oncogenic receptor, IL-17rb, in both syngeneic mouse breast tumor model and clinical patients. These results provided an unambiguous evidence to support the importance of TDLNs in promoting distant organ metastasis."

15. The authors should consult a statistician. Students t test is not appropriate for the data presented here involving two groups. Mann-Whitney U test should be applied since the data are not normally distributed.

Response: We agreed with the reviewer's comment, and re-analyzed our original Figure 6b using Mann-Whitney U test using GraphPad Prism 5 (GraphPad Software). The analysis showed a significant p-value ($P=0.0015$) and the revised results are now presented in Figure R11b and the revised manuscript.

Referee #2 (Remarks):

In this paper, Huang et al examine the role of Treg cells in draining lymph nodes (DLN) on promoting the metastatic potential of breast cancer cells. In particular, the authors propose that TRegs in DLN secrete TGFB1, leading to increased expression of IL17RB in breast cancer cells and consequently a more aggressive metastatic cancer phenotype. This paper extends a similar observation by the same authors (JEM 2015), where they proposed that increased IL17B/IL17RB signaling promotes pancreatic cancer metastasis. The study

correlates the factors influencing the *in vitro* growth characteristics of mouse mammary carcinoma 4T1 cells with the metastatic behaviour of the same 4T1 cells when transferred to mice in a syngeneic mouse breast tumor model.

Experiments are well performed and clearly presented. *In vitro* stimulation by isolated TRegs of 4T1 cell expansion is demonstrated to correlate with TGFB1 expression, TReg frequency *in vivo* in DLN, and the frequency of lung metastases was inhibited by depletion of DLN TRegs. Experiments are presented which support the hypothesis that TRegs in DLN secrete TGFB1, leading to up-regulation of IL17RB in breast cancer cells in this microenvironment and the acquisition of a more aggressive metastatic phenotype. Overall, this study provides important insight into the possible role of TRegs and the DLN micro-environment in enhancing the metastatic potential of breast cancer cells. While a direct causative link between DLN TRegs and the induction of an increased metastatic phenotype in breast cancer cells as they move through the DLN is not demonstrated, the study provides valuable data from which to examine the role of the DLN micro-environment on tumour cell maturation.

The study might be strengthened by the inclusion of experiments that further explore the functional link between TRegs, TGFB1 expression, IL17RB/IL17B signaling, and tumor metastasis.

Response: We appreciate the reviewer's encouragement and suggestions to improve our manuscript. Our aim in this study is to evaluate the promoting role of TDLN in breast cancer progression. We came upon *Il17rb* in a microarray analysis. Our previous studies on IL17RB/IL17B demonstrated its roles in tumor growth and metastasis in human breast and pancreatic cancers, respectively (Huang et al, 2014; Wu et al, 2015). We showed that IL-17B/IL-17RB signaling activated NF- κ B in human breast cancer (Huang et al, 2014). Thus, we used NF- κ B nuclear translocation as the measurement for the consequence of Il-17 induction in wild-type (WT) and Il-17rb deleted (Il-17rbDel) 4T1 cells treated with TGF- β 1. As shown in Figure R12, Il-17b recombinant protein (rIl-17b) treatment (100ng/ml) induced mild nuclear translocation of NF- κ B in the 4T1 WT cells. When 4T1 cells were pre-treated with TGF β 1 for five days, rIl-17b can trigger more nuclear translocation of NF- κ B. This phenomenon was abolished in both Il-17rbDel 4T1 cells and TGF- β 1 pre-treated Il-17rbDel 4T1 cells upon rIl-17b treatment. Thus, the results suggest that TGF- β 1 indeed can enhance Il-17rb/Il-17b signaling in Il-17rb expressing 4T1 cells, and thus contribute to the malignancy of 4T1LN cells.

Figure R12: TGF β 1 induced nuclear translocation of NF- κ B in Il-17rb expressing 4T1 cells. This data is now presented in the revised manuscript.

Our previous study also showed that IL17RB/IL17B signaling is required for pancreatic cancer metastasis and cell survival in distant organs via activating CCL20/CXCL1/IL-8/TFF1 chemokine expressions through ERK1/2 pathway (Wu et al, 2015). We tested whether the up-regulation of *Il17rb* in 4T1LN cells also activated the same pathway as in human pancreatic cancer to promote metastasis. The results showed that, possibly due to species and tissue specificity, Il17b treatment did not significantly induce ERK1/2 phosphorylation in 4T1 cells. Our microarray analysis (by KEGG pathway enrichment analysis) using 4T1PT and 4T1LN cells found that MAPK, PI3K/AKT, and Prolactin signaling could be activated in 4T1LN cells (Figure R13). Whether these pathways

were activated by Tregs-TGF β 1-IL17rb remain to be elucidated. Nevertheless, a key question in the IL17RB field is how IL-17RB expression is up-regulated in cancer cells. In this study, we provided the first mechanistic evidence showing that expression of IL-17RB in breast cancer cells can be up-regulated by Treg-secreted TGF- β 1 in the TDLN microenvironment. In the revised manuscript, we now provide more precise and reliable molecular evidence to show that TDLNs serve as an incubator for cancer cells to enhance their malignancy by Tregs-TGF β 1-IL17RB axis.

Term	P value	Genes
MAPK signaling pathway	0.033	Fgf16,Fos,Nr4a1,Sos1,Dusp1,Hspa1b
PI3K-Akt signaling pathway	0.038	Fgf16,Nr4a1,Tnxb,Sos1,Cdc37,Igga10,Thbs2
Prolactin signaling pathway	0.095	Fos, Sos1, Socs3

Figure R13: Results of KEGG pathway enrichment analysis of the up-regulated genes in 4T1LN cells.

Referee #3 (Comments on Novelty/Model System):

As clarified in my response to the authors most conclusions are drawn from 4T1 cell variants and that is the limitation.

Referee #3 (Remarks):

The authors use an established transgenic mouse model of 4T1 cells where they isolated tumor cells from lymph nodes and these displayed a more invasive/aggressive property in breast cancer (BCa) cell growth. The authors provide mechanistic insights of a key role of the IL-17R expression on BCa cells that is triggered by Treg induced TGFbeta expression acting back on the BCa cells. Their results are substantiated by xenograft transplants in immunocompetent syngenic Balb/c recipients and data is also correlated for BCa primary and secondary site metastasis human specimens with Treg analysis. The sole use of Foxp3 staining for Treg quantification on paraffin section is a limited analysis and questionable if state of the art, but the reviewer appreciates the number of paired human specimens which were pulled into the study which is certainly strength. The last experiment carried out in NSG mice lacks Tregs and is as such inconclusive. Mechanistic insights into the 4T1 cell system are interesting, but the model has questions if really reflecting a physiologic BCa model. Overall, the authors conclude that regulatory T cells in the draining lymph nodes are a key determinant of BCa progression and extravasation through their capacity to secrete TGFbeta to induce IL-17R expression in BCa cells that migrate there.

Response: We appreciate the favorable comments and suggestions from the reviewer to improve our manuscript.

We apologize if our description of the xenograft transplants of human specimens in NSG mice (Figure 7) in the original manuscript was confusing. In Figure 7, we injected the primary human breast cancer cells which freshly isolated from patient's specimens. The hBCPT cells came from the primary tumor site, and hBCLN cells came from the tumor adjacent LN metastasis. After isolation of hBCPT cells and hBCLN cells from the same patient, we mixed cells with Matrigel and injected it into immuno-compromised NSG mice respectively to analyze their tumorigenic activity.

These cancer cells were already interacted with their tumor microenvironment either at the primary site or TDLN in the patients for a long time. It is possible that the hBCLN cells in the TDLN have been "educated" long enough that the expression of IL-17RB in the hBCLN has become constantly up-regulated by epigenetic machinery. Thus, when evaluating the tumorigenic activities of the hBCPT and the hBCLN cells in mice without Tregs, hBCLN still grew faster.

IL-17 stimulates both innate immunity and host defense, where it is e.g. important for colorectal cancer progression. IL-17 plays an active role in inflammatory diseases and to

complicate the picture it is also important for autoimmune diseases. IL-17 has a key role in cancer, where multiple receptor chain forms can be expressed and two major cytokine ligands were described. The authors have published a previous manuscript (Huang et al., Oncogene, 2014) on the role of IL-17R and NF- κ B activation to be important for BCa cell survival and here they discover Tregs as the main trigger of IL-17R expression, which probably is activated.

Response: In most of the literature, IL-17 is used to refer IL-17A, and IL-17R is used to refer IL-17RA. As an important role of IL-17A/IL-17RA signaling in inflammation and autoimmune diseases, previous studies also showed that their activate role in colorectal cancer and pancreatic cancer progression (McAllister et al, 2014; Wang et al, 2014). Although IL-17RB show homology to IL-17RA and contain certain conserved structural motifs, including an extracellular fibronectin III-like domain and a cytoplasmic SEF/IL-17R (SEFIR) domain, IL-17RB did not bind IL-17A and had a TRAF6-binding motif in its cytoplasmic tail (Gaffen, 2009). Our previous study has showed IL-17RB homodimer binds IL-17B to activate downstream NF- κ B signaling via TRAF6 and thus contribute to the tumorigenesis of human breast cancer cells (Huang et al, 2014). Thus, we used NF- κ B nuclear translocation as the measurement for the consequence of IL-17b induction in wild-type and Il-17rb deleted 4T1 cells treated with TGF β 1 as described below.

Major criticism is given in the following:

1. 4T1 BCa cells are an established model, but this is not a primary BCa or GEMM (numerous genetically engineered mouse models for BCa were made where one GEMM could be evaluated). To proof in a second mouse model setting major findings of the 4T1 model is key for overall conclusion. Results for the 4T1 model are convincing as presented, but a single cell line model has limitations and might have special features that do not closely match a relevant model system for BCa. Two different cell lines or one cell line and one GEMM model based on transgenic or inbred mice seem more justified. Lentiviral integration of reporters were made and the continuous culture of 4T1 cells of unknown passages could result in unwanted superficial BCa cell system which has to be excluded by incorporation of data of a second distinct model. This limitation is the major drawback of the study. The use of a second syngenic Balb/c cell model cell lined called EMT6 is described in the paper foggy. This might be a way to go, but it can nowhere be found in the Figures as clearly marked. In general, a GEMM is superior to another long term culture cell line. The reviewer studied carefully the Figures and legends but results on EMT6 are only speculative from reading Material and Methods section that indicate it was used for the lung invasion assay. This assay is based on tail vein induction, a rather artificial assay that does not reflect physiologic metastatic spread and particular larger tumor cells stick into the blood vasculature of the lung. This has little to do with lymph vessel spreading of BCa cells and can be omitted. To substantiate major conclusions the authors need to question in a second model system in mice in an immunocompetent setting that Tregs reside significantly in the draining lymph nodes and if removed potentially increase survival.

Response: We thank reviewer's comments. We have used EMT6 cells to repeat several experiments and the results are now presented in the revised manuscript and below. As shown in Figure R14a, an increase in the prevalence of Tregs can also be observed in TDLN after EMT6 injection in Balb/c mice. We collected EMT6LN and EMT6PT cells from EMT6 tumor-bearing mice at week3 and found the expression of Il-17rb was also significantly up-regulated in EMT6 LN when compared to EMT6PT cells (Figure R14b). EMT6LN cells also showed higher abilities in soft-agar colony formation, tumor growth, and lung colonization than EMT6PT cells (Figure R14c-g). The data is now presented in the Supplementary Figure 2 of the revised manuscript.

Figure R14: EMT6 cells isolated from tumor-draining lymph nodes showed increased expression of Il-17rb and enhanced malignancy. Tregs are increased in EMT6 tumor-bearing mice (a). Compared to EMT6_{PT} cells, EMT6_{LN} cells have an increased Il-17rb up-regulation (b), soft-agar colony forming ability (c), tumor growth ability (d and e), and lung colonization ability (f and g). The data is now presented in the revised manuscript.

We apologized for the inappropriate interpretation for the results from the tail vein injection in the original manuscript. To further demonstrate that 4T1LN cells have greater metastatic potential than 4T1PT cells, we isolated and injected 4T1LN, 4T1PT, EMT6LN and EMT6PT cells into both side of the 4th mammary fat pads of NSG mice. We monitored the distant organ metastasis by IVIS imaging. As shown in Figure R7, a higher percentage of distant organ metastasis in 4T1LN- or EMT6LN-injected NSG mice compared to 4T1PT- or EMT6PT-injected NSG mice was observed. Thus, this new results provide evidence that 4T1LN cells have greater metastatic potential than 4T1PT cells. This new data is presented in the figure 1H-II of the revised manuscript.

Figure R7: 4T1LN cells and EMT6LN cells showed greater metastatic potential than 4T1PT and EMT6PT cells. This data is now presented in the revised manuscript.

We believe, with the evidence from both Figure R7 and R14, that TDLNs can enhance malignancy of breast cancer cells via up-regulating IL-17rb is not limited to 4T1 cells alone.

2. IL-17 is largely produced by activated memory T lymphocytes. CD8⁺ memory effector cells or activated T cells in general express CD25 (B cells or malignant myeloid progenitors can express CD25), which they require for growth and survival. Thus, the use of CD25 antibody injection for Treg ablation has to be described with more care; certainly a genetic deletion of Tregs by knockout of *Foxp3* in the T cell lineage e.g. is superior. Moreover, IL-17 can also act on Th17 cells that are also CD4⁺. Therefore, activated memory effector CD8⁺ T cells and Th17 cells should be quantified in parallel to Treg quantifications as described in results to Figure 4. Treg or Th17 cells can be grown in vitro similarly after TCR and costimulatory receptor engagement of purified naïve CD4⁺/CD62L⁺ subsets and both populations expand with stimulation of TGFβ1. Th17 cells require both TGFβ1 and IL-6. To detect both subsets intracellular FACS for *Foxp3* and IL-17 or *Foxp3* and CD25 could be done. Th17 cells or CD8⁺ T cells cannot be excluded and could be analyzed in Figure 4.

Response: We thank the reviewer for pointing out that anti-CD25 antibody (Ab) injection depletes not only Tregs, but also activated CD4⁺ T cells, activated CD8⁺ T cells, and Bregs. Bregs has also been reported involving in lung metastasis in a 4T1 mouse model (Olkhanud et al, 2011). We agree with the reviewer that the study will be better if we used *Foxp3*^{-/-} mice. Although we do not have such mouse strain for the experiment, we employed additional experiments, which are also suggested by the reviewer 1, to strengthen our conclusion.

As shown in the original Figure 4a-4d, we sorted different immune cell subsets by FACS for the transwell co-culture assay with 4T1 cells. These results showed that only CD4⁺CD25⁺ T cells have the ability to induce the expression of IL-17rb in 4T1 cells. We then used anti-CD4 Ab injection (includes all *Foxp3*⁺ Tregs) at TDLN in 4T1 tumor-bearing mice to depleted all CD4⁺ T cells but not Bregs (Figure R15). We also used anti-CD25 Ab injection as a control. As shown in the Figure R9, the expression of *Il-17rb* in the 4T1LN cells collected from either CD4⁺ T cells-depleted mice or CD25⁺ cells-depleted mice was significantly abolished (Figure R9b). Moreover, both tumor growth and distant organ metastasis (IVIS image assay were taken on day 21) in 4T1LN from the CD4⁺ T cells-depleted mice or CD25⁺ cells-depleted mice were significantly reduced (Figure R9c-e). Thus, these results indicate that CD4⁺CD25⁺ T cells are the dominant effectors. Furthermore,

our original Figure 5a showed that the neutralization of TGF- β 1 by anti-TGF- β 1 can abolish Il-17rb expression on 4T1 cells in a transwell co-culture assay with CD4+CD25+ T cells. It is known that TGF β 1 is one of the secreted cytokines from Foxp3+ Tregs but not from other CD4+ T cell subsets (Vignali et al, 2008). Therefore, these results suggest that Tregs in TDLN is essential for Il-17rb expression in breast cancer cells to enhance cancer malignancy. These data are now presented in the Figure 4E-4H, Figure 5, Figure 6A, and Supplementary Figure 6 of the revised manuscript.

Original Figure 4A-4D: CD4+CD25+ cells in the tumor-draining lymph node microenvironment are required for Il-17rb induction in breast cancer cells.

Figure R15: 4T1-injected BALB/c mice were treated with control IgG (a), anti-CD4 (b), or anti-CD25 (c) neutralizing antibody at the day 15 and day 18 after initial injection. Mice were sacrificed at three weeks after initial injection. The CD4+CD25+ Treg population (Left panel), and

B220+CD25+ Breg (Right panel) population in inguinal LNs were confirmed by FACS analysis. These data are now presented in the revised manuscript.

Figure R9: Depleting total CD4⁺ T cells or CD25⁺ cells in the tumor draining LN (a) abolished Il-17rb induction (b), tumor growth ability (c and d), and metastatic activity of 4T1LN cells (e). These data are now presented in the revised manuscript.

We screened the percentage of the memory effector CD8 T cells at the TDLN in the 4T1 tumor-bearing mice. As shown in Figure R16, in comparison with CD4⁺Foxp3⁺ Tregs, we did not find a significant increase in CD44⁺CD62⁻ memory effector CD8⁺ T cells. Moreover, *IL-17rb* expression was significantly induced when 4T1 cells were co-cultured only with CD4⁺ T cell subset, but not with CD8⁺ T cell subset as shown in the original Figure 4a-4b. Since Th17 cells were increased in the blood, spleen and primary tumor in 4T1-injected mice (Qian et al, 2013), the population of ROR γ t⁺ Th17 cells or IL-17A⁺ CD4⁺ T cells in TDLNs were also examined by FACS analysis. However, in comparison with CD4⁺Foxp3⁺ Tregs, no significant increase in ROR γ t⁺ Th17 cells or IL-17A⁺ CD4⁺ T cells was found in the TDLNs in the third week after 4T1 cell injection (Figure R16). Thus, it is likely that Tregs in the TDLNs may be responsible for the induction of *IL-17rb* in cancer cells. These results are now presented in the Supplementary Figure 5 of the revised manuscript.

Figure R16: The dynamic composition of Tregs (upper left), ROR γ t⁺ Th17 cells (upper right), IL-17A⁺ Th17 cells (lower left), and CD8⁺ effector memory T cells (lower right) in the tumor-draining LNs after 4T1 cell injection. These data are now presented in the revised manuscript.

3. IL-17 signaling can activate different transcription factors (e.g. AP-1, NF κ B or C/EBP family members) which then might also change their expression levels. The activation of the IL-17R in BCa cells is likely in their 4T1 model system, but it would be important to show it e.g. in the paired 4T1 cells with and without Th17R expression (using the CRISP-Cas9 clone) after pre-stimulation with TGF β 1 to visualize consequences of IL-17 induction on downstream transcription factor action.

Response: Our previous study showed that IL-17B/IL-17RB signaling activated NF- κ B in human breast cancer (Huang et al, 2014). Thus, we used NF- κ B nuclear translocation as the measurement for the consequence of IL-17 induction in wild-type (wt) and IL-17rb deleted (IL-17rbDel) 4T1 cells treated with TGF β 1. As shown in Figure R12, IL-17b recombinant protein (rIL-17b) treatment (100ng/ml) induced mild nuclear translocation of NF- κ B in the 4T1 wt cells. When 4T1 cells were pre-treated with TGF β 1 for five days, rIL-17b can trigger more nuclear translocation of NF- κ B. This phenomenon was abolished in both IL-17rbDel 4T1 cells and TGF β 1 pre-treated IL-17rbDel 4T1 cells upon rIL-17b treatment. Thus, the results suggest that TGF β 1 indeed can enhance IL-17rb/IL-17b

signaling in IL-17rb expressing 4T1 cells, and thus contribute to the malignancy of 4T1LN cells. This data is now presented in the Figure 6I-6J of the revised manuscript.

Figure R12: TGFβ1 induced nuclear translocation of NF-κB in IL-17rb expressing 4T1 cells. This data is now presented in the revised manuscript.

Minor:

4. It is questionable if a functional IL-17 response is triggered in the 4T1 cell pairs. Is expression of nuclear factor-kappa B, chemokines CXCL8, CXCL6 and CXCL1 or myeloid cytokines such G-CSF and GM-CSF, the inflammatory cytokine IL-6 or the adhesion molecule ICAM-1 that were all described to be induced upon IL-17 stimulation distinct in the paired 4T1 cells could add more mechanistic data. A comprehensive microarray would answer that, but focused real time PCR could be done. The authors should provide complete data with significantly up- and down-regulated gene expression in e.g. Suppl. Table format on the microarray mRNA expression analysis and not only show a selected set of five surface marker gene expressions, since that analysis looks narrow or biased. If these data have been generated new it should be described as such. However, if it is from previous work, which was from reading as described unclear, it should be marked as such.

Response: We agree with the reviewer's concern. Our cDNA microarray raw data have been deposited in the Gene Expression Omnibus (GEO) repository (GEO: GSE86971). Genes with a log₂ 1.5-fold differential expression between 4T1PT and 4T1LN cells were listed in Supplementary Table R1 and R2 of revised manuscript. To explore the possible signaling pathway activated in 4T1LN cells, we used DAVID (The Database for Annotation, Visualization and Integrated Discovery) Bioinformatics Resources 6.8 (available at <https://david.ncifcrf.gov/home.jsp>) to carry out the Kyoto Encyclopedia of Genes and Genomes (KEGG) pathway annotation for the up-regulated genes. $P < 0.1$ was selected as the cutoff criterion. In a KEGG pathway enrichment analysis of all up-regulated 143 genes, we found that MAPK, PI3K/AKT, and Prolactin signaling pathway were significantly enriched (Figure R13). MAPK (Whyte et al, 2009) and PI3K/AKT (Dillon et al, 2007) signaling pathways have well-established roles in breast cancer progression. Interestingly, we found that several cytokine and chemokine genes including *Il-1a*, *Il-6*, *Il-23a*, *csf-3*, *ccl2*, *ccl3*, *ccl7*, *ccl20*, *cxcl2*, and *cxcl5* were significantly down-regulated in 4T1LN cells (Figure R17). It is possible that many cross-interactions can occur between cancer cells and immune cells besides the cancer cells and Tregs in the TDLN microenvironment and therefore resulting in the down-regulation of these cytokine genes. To elucidate whether MAPK, PI3K/AKT, and Prolactin signaling pathway correlate with IL-17B/IL-17RB downstream signaling in 4T1LN cells, we compare this result with our previous cDNA microarray raw data of IL-17RB-depleted MDA-MB-361 cells (Huang et al, 2014). In a KEGG pathway enrichment analysis of down-regulated genes in IL-17RB-depleted MDA-MB-361 cells ($P < 0.1$ was selected as the cutoff criterion), we found that MAPK signaling pathway was also enriched (Figure R18). Moreover, we used RT-qPCR assay to confirm that these genes of MAPK pathway were up-regulated in 4T1LN cells (Figure R19). However, it remains to be investigated whether this group of genes is directly regulated through the TGF-β1 pathway or IL-17B/IL-17RB pathway or both. Nevertheless, the activation of these genes in 4T1LN cells is consistent with the notion of enhanced cancer malignancy. These data are now presented in the Supplementary Table 4, Supplementary Table 5, and Supplementary Figure 10 of the revised manuscript.

Term	P value	Genes
MAPK signaling pathway	0.033	Fgf16,Fos,Nr4a1,Sos1,Dusp1,Hspa1b
PI3K-Akt signaling pathway	0.038	Fgf16,Nr4a1,Tnxb,Sos1,Cdc37,Itga10,Thbs2
Prolactin signaling pathway	0.095	Fos, Sos1, Socs3

Figure R13: Results of KEGG pathway enrichment analysis of the up-regulated genes in 4T1LN cells. This data is now presented in the revised manuscript.

Gene	4T1 _{LN} /4T1 _{PT} (log ₂)
Ccl2	-5.52
Ccl20	-3.89
Il-1a	-2.98
Il-24	-2.69
Cxcl5	-2.31
Ccl7	-2.03
Cxcl3	-1.9
Csf-3	-1.79
Il-23a	-1.75
Il-6	-1.74

Figure R17: The down-regulated cytokine and chemokine genes in 4T1LN cells.

Term	P value	Genes
Transcriptional misregulation in cancer	0.073	DDIT3,ETV4,SUPT3H,ARNT2,MMP9
Pathways in cancer	0.075	MECOM,WNT3,ARNT2,FGF1,LAMA2,MMP1, MMP9,PDGFRB
MAPK signaling pathway	0.091	DDOT3,MECOM,DUSP4,FGF1,PDGFRB,PTPRR
Staphylococcus aureus infection	0.091	C1R,C5,HLA-DRA
Legionellosis	0.091	CXCL2,CXCL3,EEF1A2

Figure R18: Results of KEGG pathway enrichment analysis of the down-regulated genes in shIL-17RB MDA-MB-361 cells. This data is now presented in the revised manuscript.

Figure R19: mRNA expressions of *Fos*, *Nr4a1*, *Sos1*, *Dusp1*, *Hspa1b* genes in 4T1PT and 4T1LN cells were determined by RT-qPCR. This data is now presented in the revised manuscript.

5. The ChIP data in Figure 5h suggest that SMAD proteins are continuously bound at the IL-17rb locus, but SMAD proteins are activated by serine/threonine phosphorylation upon TGFbeta stimulation, where they efficiently shuttle to the nucleus. Therefore, the ChIP is

unclear and a better analysis with Western blot control on several SMAD family member expressions in BCa cells upon TGFbeta stimulation could clarify if indeed pSMAD4 is significantly expressed. Nuclear staining of pSMAD4 could be shown and quantified in respect to another SMAD family member control, protein analysis of 4T1 extracts prior to ChIP by immunoblotting with specific phospho- and total protein antibodies then would allow a better interpretation of transcription factors driving IL-17R expression, which can be driven also by other transcription factors as reasoned above in major point 2.

Response: Our original ChIP results showed that an increased binding of Smad4 on the Il-17rb promoter can be found upon TGF- β 1 treatment. To further demonstrate that TGF- β 1 signaling results in transcriptional up-regulation of Il17rb, we first used immuno-fluorescence staining to examine whether the nuclear staining of pSMAD4 is increased after TGF β stimulation. However, we found the strong nuclear staining of pSMAD4 in 4T1 cells without any treatment. There was no difference between un-treated and TGF- β 1-treated 4T1 cells (Figure R20). Thus, we used Western blot to analyze the sub-cellular localization of SMADs in 4T1 cells upon TGF- β 1 treatment. We used commercial sub-cellular protein fractionation kit for cultured cells (Thermo Scientific #78840) to isolated cytoplasm-, nucleoplasm-, and chromatin binding-fractions of TGF- β 1-treated 4T1 cells. As shown in Figure R21, TGF- β 1 treatment induced the chromatin binding of Smad2/3 and Smad4 molecules, but not the inhibitory Smad6 molecule. We then tested which Smad molecule was necessary for Il-17rb expression. We used lentiviral to deliver Smad2, Smad3 or Smad4 shRNA into 4T1 cell and examined the Il-17rb induction after TGF- β 1 treatment. The knockdown of individual Smad molecule was specific and is confirmed by Western blot assay (Figure R22a). Surprisingly, TGF- β 1-induced Il-17rb expression was inhibited not only by Smad4 depletion, but also by Smad2 or Smad3 depletion (Figure R22b-d). It is known that TGF- β 1 signaling results in Smad2-Smad4-cofactor or Smad3-Smad4-cofactor complex formation in the nucleus to regulate target gene expression (Massague, 2008). Which cofactor in these complexes to regulate Il-17rb transcription still needs to be elucidated. Nevertheless, these new results illustrate the importance of TGF- β 1 downstream Smad2, Smad3, and Smad4 signaling in Il-17rb transcriptional regulation. These results are now presented in the Figure 6F-6H and Supplementary Figure 8 of the revised manuscript.

Figure R20: The nuclear staining of pSMAD4 in 4T1 cells.

Figure R21: TGFβ treatment induces the chromatin binding of Smad2, Smad3, and Smad4 in 4T1 cells. This data is now presented in the revised manuscript.

Figure R22: TGF β -mediated Il-17rb up-regulation is abolished by (b) Smad2-, (c) Smad3-, or (d) Smad4 depletion in 4T1 cells. (a) The knockdown of individual Smad molecule is specific. These results are now presented in the revised manuscript.

6. Figure 3e is unclear why not genetic deletion of the clone with CRISP-Cas9 was used as well which could be superior to the shRNA knockdown of 17rb. One reason might be controls, but at least a similar experiment could be performed since the shRNA 4T1 derivatives provide controls and result could be more meaningful.

Response: We agree with the reviewer's concern. The Il-17rbDel 4T1 cells are superior to the shIl-17rb 4T1 cells. We have performed the similar experiment using Il-17rbDel 4T1 cells in the transwell co-culture with lymphocytes isolated from WT LN or TDLNs. As shown in Figure R23, we found that the colony forming ability was not induced in both shIl-17rb 4T1 cells and Il-17rbDel 4T1 cells when compared to shLacZ 4T1 cells and WT 4T1 cells. However, the significant reduction of colony forming ability was found in Il-17rbDel 4T1 cells when co-culture with lymphocytes isolated from TDLNs. This phenomenon may due to the regulation of survival signaling via IL-17B/IL-17RB activated NF- κ B in the human breast cancer cells in our previous study (Huang et al, 2014). We also showed that the induction of NF- κ B nuclear translocation by rIl-17b was abolished in Il-17rbDel 4T1 cells (Figure R12). Thus, this result illustrated that IL-17B/IL-17RB response can contribute to survival of 4T1LN cells.

Figure R23: Depletion of *Il-17rb* by using either shRNA-mediated knockdown (a) or CRISP-Cas9-mediated knockout (b) abolishes the colony forming ability induced by the tumor-draining LNs.

Reference:

- Dillon RL, White DE, Muller WJ (2007) The phosphatidyl inositol 3-kinase signaling network: implications for human breast cancer. *Oncogene* **26**(9): 1338-1345
- Gaffen SL (2009) Structure and signalling in the IL-17 receptor family. *Nat Rev Immunol* **9**(8): 556-567
- Gao H, Chakraborty G, Lee-Lim AP, Mo Q, Decker M, Vonica A, Shen R, Brogi E, Brivanlou AH, Giancotti FG (2012) The BMP inhibitor Coco reactivates breast cancer cells at lung metastatic sites. *Cell* **150**(4): 764-779
- Gervasoni JE, Jr., Sbayi S, Cady B (2007) Role of lymphadenectomy in surgical treatment of solid tumors: an update on the clinical data. *Ann Surg Oncol* **14**(9): 2443-2462
- Hirakawa S, Brown LF, Kodama S, Paavonen K, Alitalo K, Detmar M (2007) VEGF-C-induced lymphangiogenesis in sentinel lymph nodes promotes tumor metastasis to distant sites. *Blood* **109**(3): 1010-1017
- Huang CK, Yang CY, Jeng YM, Chen CL, Wu HH, Chang YC, Ma C, Kuo WH, Chang KJ, Shew JY, Lee WH (2014) Autocrine/paracrine mechanism of interleukin-17B receptor promotes breast tumorigenesis through NF-kappaB-mediated antiapoptotic pathway. *Oncogene* **33**(23): 2968-2977
- Massague J (2008) TGFbeta in Cancer. *Cell* **134**(2): 215-230
- McAllister F, Bailey JM, Alsina J, Nirschl CJ, Sharma R, Fan H, Rattigan Y, Roeser JC, Lankapalli RH, Zhang H, Jaffee EM, Drake CG, Housseau F, Maitra A, Kolls JK, Sears CL, Pardoll DM, Leach SD (2014) Oncogenic Kras activates a hematopoietic-to-epithelial IL-17 signaling axis in preinvasive pancreatic neoplasia. *Cancer Cell* **25**(5): 621-637
- Olkhanud PB, Damdinsuren B, Bodogai M, Gress RE, Sen R, Wejksza K, Malchinkhuu E, Wersto RP, Biragyn A (2011) Tumor-evoked regulatory B cells promote breast cancer metastasis by converting resting CD4(+) T cells to T-regulatory cells. *Cancer Res* **71**(10): 3505-3515
- Ouzounova M, Lee E, Piranlioglu R, El Andaloussi A, Kolhe R, Demirci MF, Marasco D, Asm I, Chadli A, Hassan KA, Thangaraju M, Zhou G, Arbab AS, Cowell JK, Korkaya H (2017) Monocytic and granulocytic myeloid derived suppressor cells differentially regulate spatiotemporal tumour plasticity during metastatic cascade. *Nat Commun* **8**: 14979
- Pereira ER, Jones D, Jung K, Padera TP (2015) The lymph node microenvironment and its role in the progression of metastatic cancer. *Semin Cell Dev Biol* **38**: 98-105
- Qian X, Gu L, Ning H, Zhang Y, Hsueh EC, Fu M, Hu X, Wei L, Hoft DF, Liu J (2013) Increased Th17 cells in the tumor microenvironment is mediated by IL-23 via tumor-secreted prostaglandin E2. *J Immunol* **190**(11): 5894-5902
- Ran S, Volk L, Hall K, Flister MJ (2010) Lymphangiogenesis and lymphatic metastasis in breast cancer. *Pathophysiology* **17**(4): 229-251
- Rouzier R, Extra JM, Klijanienko J, Falcou MC, Asselain B, Vincent-Salomon A, Vielh P, Bourstyn E (2002) Incidence and prognostic significance of complete axillary downstaging after primary chemotherapy in breast cancer patients with T1 to T3 tumors and cytologically proven axillary metastatic lymph nodes. *J Clin Oncol* **20**(5): 1304-1310
- Shibata MA, Morimoto J, Shibata E, Otsuki Y (2008) Combination therapy with short interfering RNA vectors against VEGF-C and VEGF-A suppresses lymph node and lung metastasis in a mouse immunocompetent mammary cancer model. *Cancer Gene Ther* **15**(12): 776-786
- Tyan SW, Hsu CH, Peng KL, Chen CC, Kuo WH, Lee EY, Shew JY, Chang KJ, Juan LJ, Lee WH (2012) Breast cancer cells induce stromal fibroblasts to secrete ADAMTS1 for cancer invasion through an epigenetic change. *PLoS One* **7**(4): e35128

Tyan SW, Kuo WH, Huang CK, Pan CC, Shew JY, Chang KJ, Lee EY, Lee WH (2011) Breast cancer cells induce cancer-associated fibroblasts to secrete hepatocyte growth factor to enhance breast tumorigenesis. *PLoS One* **6**(1): e15313

Vignali DA, Collison LW, Workman CJ (2008) How regulatory T cells work. *Nat Rev Immunol* **8**(7): 523-532

Wang K, Kim MK, Di Caro G, Wong J, Shalpour S, Wan J, Zhang W, Zhong Z, Sanchez-Lopez E, Wu LW, Taniguchi K, Feng Y, Fearon E, Grivennikov SI, Karin M (2014) Interleukin-17 receptor signaling in transformed enterocytes promotes early colorectal tumorigenesis. *Immunity* **41**(6): 1052-1063

Wang Y, Klijn JG, Zhang Y, Sieuwerts AM, Look MP, Yang F, Talantov D, Timmermans M, Meijer-van Gelder ME, Yu J, Jatkoa T, Berns EM, Atkins D, Foekens JA (2005) Gene-expression profiles to predict distant metastasis of lymph-node-negative primary breast cancer. *Lancet* **365**(9460): 671-679

Weigelt B, Peterse JL, van 't Veer LJ (2005) Breast cancer metastasis: markers and models. *Nat Rev Cancer* **5**(8): 591-602

Whyte J, Bergin O, Bianchi A, McNally S, Martin F (2009) Key signalling nodes in mammary gland development and cancer. Mitogen-activated protein kinase signalling in experimental models of breast cancer progression and in mammary gland development. *Breast Cancer Res* **11**(5): 209
Wu HH, Hwang-Verslues WW, Lee WH, Huang CK, Wei PC, Chen CL, Shew JY, Lee EY, Jeng YM, Tien YW, Ma C (2015) Targeting IL-17B-IL-17RB signaling with an anti-IL-17RB antibody blocks pancreatic cancer metastasis by silencing multiple chemokines. *J Exp Med* **212**(3): 333-349

2nd Editorial Decision

30 August 2017

Thank you for the submission of your revised manuscript to EMBO Molecular Medicine. We have now received the enclosed reports from the reviewers that were asked to re-assess it. As you will see the reviewers are now supportive, although reviewer #1 has a few final suggestions, with which we agree, that require your action.

I am thus prepared to accept your manuscript for publication pending satisfactory compliance with the reviewer's final requests. Please also carefully fulfil the following editorial requirements:

- 1) The conflict of interest statement is missing from the manuscript file.
- 2) Please upload the manuscript as a .doc file (not .pdf)
- 3) You have provided a number of supplementary figures and tables. Please note that up to 5 supplementary figures plus a number of tables can be chosen for inclusion in the article as Expanded View figures, which must then be uploaded individually. The remaining figures and tables should be included in an Appendix to be provided as a PDF file. The Appendix should begin with a short table of contents (not "Inventory of Supplementary information"). As a consequence, the manuscript callouts and legends for all supplementary figures and tables (EV and Appendix) will have to be carefully amended where necessary to reflect the correct nomenclature: Expanded view figures and tables are referred to in the text as Figure EV1, Table EV1, etc. Appendix figures and tables as Appendix Figure S1, Appendix Table S1, etc. For all the above, please refer to our detailed author guidelines (embomolmed.embopress.org/authorguide#expandedview).
- 4) Although after your revision the callouts will change significantly, I do wish to mention that some of the figures are not currently referred to in the manuscript text. Please pay special attention to this.
- 5) We note that you provide source data. This is much appreciated, thank you. However, please provide one PDF file per each figure and upload separately as source data files, not as supplementary information. The files should be labeled with the appropriate figure/panel number,

and in the case of gels, should have molecular weight markers; further annotation may be useful but is not essential. The files will be published online with the article as supplementary "Source Data" files.

6) As per our Author Guidelines, the description of all reported data that includes statistical testing must state the name of the statistical test used to generate error bars and P values, the number (n) of independent experiments underlying each data point (not replicate measures of one sample), and the actual P value for each test (not merely 'significant' or ' $P < 0.05$ ').

7) Please note that EMBO Molecular Medicine now requires a complete author checklist (<http://embomolmed.embopress.org/authorguide#editorial3>) to be submitted with all revised manuscripts. Provision of the author checklist is mandatory at revision stage; The checklist is designed to enhance and standardize reporting of key information in research papers and to support reanalysis and repetition of experiments by the community. The list covers key information for figure panels and captions and focuses on statistics, the reporting of reagents, animal models and human subject-derived data, as well as guidance to optimise data accessibility. The Author checklist will be published alongside the paper, in case of acceptance, within the transparent review process file.

8) Every published paper includes a 'Synopsis' to further enhance discoverability. Synopses are displayed on the journal webpage and are freely accessible to all readers. They include a short description as well as 2-5 one-sentence bullet points that summarise the key NEW findings of the paper. The bullet points should be designed to be complementary to the abstract - i.e. not repeat the same text. We encourage inclusion of key acronyms and quantitative information. Please use the passive voice. Please attach this information in a separate file or send them by email, we will incorporate it accordingly. We also encourage the provision of striking image or visual abstract to illustrate your article. If you do, please provide a jpeg file 550 px-wide x 400-px high.

9) We now mandate that all corresponding authors list an ORCID digital identifier. You may acquire one through our web platform upon submission and the procedure takes <90 seconds to complete. We also encourage co-authors to supply an ORCID identifier, which will be linked to their name for unambiguous name identification.

10) Please make sure that in your next revision, the figures and source data images are provided at a much better quality than they are currently are.

11) Lastly, I am proposing some minor modifications to the title and abstract with the aim to improve clarity and impact. Please find attached a .doc file with my proposal. Please go over it carefully and, if you agree, please use them in your final manuscript (or otherwise modify further).

Please submit your revised manuscript within two weeks. I look forward to seeing a revised form of your manuscript as soon as possible.

***** Reviewer's comments *****

Referee #1 (Comments on Novelty/Model System for Author):

The mechanism has been thoroughly tested, but the model system could be better with the use of GEMM.

Referee #1 (Remarks for Author):

The manuscript has greatly improved with further experimentation and clarification of the text. I am still unsure about how to interpret the experiments where the lymph node has been removed. It could be that Tregs are educated in the lymph nodes and subsequently migrate to other organs to suppress anti-tumor immunity. I doubt that cancer cells in lymph nodes could make their way to lungs considering the fact that metastasis is such an inefficient process. Nonetheless, the study is interesting, and now convincing. I would only ask two more things: 1) The authors must include methodology on how they generated the LM-EMT6 sub clone. It is imperative that the reader be

informed of this; and 2) I may have overlooked but I cannot find Figure R8 (as presented in rebuttal letter) in the manuscript. Please include as supplementary or main figures.

Referee #3 (Comments on Novelty/Model System for Author):

The authors invested in the revised manuscript much more work to justify their major conclusions and thus, the study has now sufficient novelty, technical quality is excellent and it is of high medical impact, since breast cancer metastasis is a devastating problem and their insights on IL-17 and TGF-beta signaling connection are important.

Referee #3 (Remarks for Author):

The reviewer read carefully the point to point reply and the revised manuscript new data and Figure additions. The conclusion is that the authors were able to recapitulate major findings also with a second independent cell line model system. In addition, they did a wealth of experiments during revision work to justify their major conclusions or to control technically their study. Also the edited text changes and overstatement removal is now well suited for acceptance and one can congratulate the team for a hard revision work.

2nd Revision - authors' response

08 September 2017

Detailed point-by-point response to reviewers' comments (bold)

Referee #1 (Comments on Novelty/Model System for Author):

The mechanism has been thoroughly tested, but the model system could be better with the use of GEMM.

Referee #1 (Remarks for Author):

The manuscript has greatly improved with further experimentation and clarification of the text. Nonetheless, the study is interesting, and now convincing. I would only ask two more things: 1) The authors must include methodology on how they generated the LM-EMT6 subclone. It is imperative that the reader be informed of this.

Response: Thank you for the positive comments. We added a more detailed methodology as described below:

Generation of EMT6 subclone from lung metastasis (LM-EMT6) in EMT6 tumor-bearing mice: To generate the LM-EMT6 subclone from allograft, mice were sacrificed at the ten weeks post 1×10^5 EMT6 cells injection in the 4th mammary fat pad. Lung tissues were minced and digested with collagenase type I (200 U/ml) and hyaluronidase (50 U/ml) in the serum free DMEM/F12 medium for 16 hr in a humidified 37°C incubator supplemented with 5% CO₂. After digestion, cell suspensions were filtrated through an 40 µm cell strainers (BD Biosciences, San Jose, CA, USA). To enrich homogenous LM-EMT6 cells, cells were incubated with APC-conjugated rat anti-mouse CD24 (#101814, Biolegend, San Diego, CA, USA) and PE-conjugated hamster anti-mouse CD29 antibodies (#102208, Biolegend, San Diego, CA, USA) for 30 mins at 4°C. The CD24⁺CD29⁺ population (Gao et al, 2012) was sorted using a FACS Aria II cell sorter (BD Bioscience, San Jose, CA, USA). After sorting, LM-EMT6 cells were cultured in DMEM/F12 medium supplemented with 10% FBS. LM-EMT6 cells transduced with green fluorescent protein (GFP) and luciferase by lentivirus were used for the allograft assays and the following surgical removal of primary tumors and TDLN assays.

2) I may have overlooked but I cannot find Figure R8 (as presented in rebuttal letter) in the manuscript. Please include as supplementary or main figures.

Response: We have added figure R8 in the Appendix Figure S3 of the revised manuscript.

Referee #3 (Comments on Novelty/Model System for Author):

The authors invested in the revised manuscript much more work to justify their major conclusions and thus, the study has now sufficient novelty, technical quality is excellent and it is of high medical impact, since breast cancer metastasis is a devastating problem and their insights on IL-17 and TGF-beta signaling connection are important.

Reply: Thank you very much!

Referee #3 (Remarks for Author):

The reviewer read carefully the point to point reply and the revised manuscript new data and Figure additions. The conclusion is that the authors were able to recapitulate major findings also with a second independent cell line model system. In addition, they did a wealth of experiments during revision work to justify their major conclusions or to control technically their study. Also the edited text changes and overstatement removal is now well suited for acceptance and one can congratulate the team for a hard revision work.

Response: We appreciate the reviewer's encouragement.

Corresponding Author Name: Dr. Wen-Hwa Lee

Manuscript Number: EMM-2016-06914